



# Investigating the Relationship between Simulation Parameters and Flow Variables in Simulating Atmospheric Gravity Waves for Wind Energy Applications

Mehtab Ahmed Khan[1], Dries Allaerts[1,†], Simon J. Watson[1], and Matthew J. Churchfield[2]

[1]Delft University of Technology, Netherlands
[†]deceased, 10 Oct 2024
[2]National Renewable Energy Laboratory, USA

**Correspondence:** Mehtab Ahmed Khan (m.a.khan-2@tudelft.nl)

**Abstract.** Wind farms, particularly offshore clusters, are becoming larger than ever before. Besides influencing the surface wind flow and the inflow for downstream wind farms, large wind farms can trigger atmospheric gravity waves in the inversion layer and the free atmosphere aloft. Wind farm-induced gravity waves can cause adverse pressure gradients upstream of the wind farm, that contribute to the global blockage effect, and can induce favorable pressure gradients above and downstream of the wind farm that enhance wake recovery. Numerical modeling is a powerful means of studying these wind farm-induced atmospheric gravity waves, but it comes with the challenge of handling spurious reflections of these waves from domain boundaries. Typically, approaches which employ radiation boundary conditions and forcing zones are used to avoid these reflections. However, the simulation setup of these approaches heavily relies on ad-hoc processes. For instance, the widely used Rayleigh damping method requires ad-hoc tuning to produce a setup which may be only produce satisfactory results for a particular case. To provide more systematic guidance on setting up realistic simulations of atmospheric gravity waves, we conduct an LES study of flow over a 2D hill and through a wind farm canopy that explores the optimum domain size and damping layer setup depending on the fundamental parameters which determine the flow characteristics.

In this work, we only consider linearly stratified conditions (i.e., no inversion layer), thereby focusing on internal gravity waves in the free atmosphere and their reflections from the domain boundaries. This type of flow is governed by a single Froude number, which dictates most of the internal wave properties, such as wavelength, amplitude, and direction. This in turn will dictate the optimum domain size and Rayleigh damping layer setup. We find the effective horizontal and vertical wavelengths, (the representative wavelengths of the entire wave spectrum), to be the appropriate length scales to size the domain and damping layer thickness, and the optimal Rayleigh damping coefficient scales with the Brunt-Väisälä frequency.

Considering Froude numbers seen in wind farm applications, we propose recommendations to limit the reflections to less than $10\%$ of the total upwards propagating wave energy. Typically, damping is done at the top boundary, but given the non-periodic lateral boundary conditions of practical wind farm simulation domains, we find that damping the inflow-outflow boundaries is of equal importance to the top boundary. The Brunt-Väisälä frequency-normalized damping coefficient should be between 1 and 10. The damping layer thickness should be at least one effective vertical wavelength; damping layers exceeding 1.5 times the vertical wavelength are found to be unnecessary. The domain length and height should accommodate at least one



effective horizontal and vertical wavelength, respectively. Moreover, Rayleigh damping does not damp the waves completely, and the non-damped energy might accumulate over the simulation time.

# 1   Introduction

The size of a modern wind farm, especially offshore, can extend several tens of kilometers horizontally, involving flow interactions on a regional scale with impacts well into the free atmosphere. The energy and momentum extraction caused by a large
wind farm is significant enough to decelerate the flow in the atmospheric boundary layer (ABL) (Frandsen, 1992; Calaf et al., 2010; Smith, 2010), which slowly recovers because of turbulent momentum transfer and the interplay of the large-scale driving pressure gradient and the Coriolis force. This is evident from the consolidated wake of a large wind farm, which extends far beyond the wind farm in the streamwise direction. Researchers frequently study the behavior of a wind farm within the ABL only and do not consider the free atmosphere above. However, a full understanding of large wind farm behavior requires an
understanding of the farm's effect on the free atmosphere and vice versa.

The temperature stratification of both the ABL and the free atmosphere is strongly connected to the flow dynamics of large wind farms. For instance, the thickness of the ABL can significantly decrease during stable atmospheric conditions, which will affect the flow at turbine level in a variety of ways. In offshore environments in particular, the height of the capping inversion (the relatively thin strongly stable layer that often forms between the top of the ABL and the free atmosphere) can drop to a
few hundred meters at night. In such conditions, wind farms can induce atmospheric gravity waves (AGWs), which include trapped gravity waves (TGWs) in the capping inversion and internal gravity waves (IGWs) in the free atmosphere aloft. Wind farm-induced AGWs were first hypothesized by Smith (2010), who treated wind farms as semi-permeable obstacles that can deflect the flow upwards to displace the capping inversion, resulting in these buoyancy-driven waves.

Wind farm-induced AGWs can affect wind farm performance by creating streamwise pressure gradients that accelerate or
decelerate the flow into and through the farm, contributing to the phenomenon of "wind farm blockage", and affecting the way the consolidated wind farm wake recovers. Smith (2010), Allaerts et al. (2018), and Lanzilao and Meyers (2021) believe that AGWs are the main driver behind the wind farm blockage effect at the entrance of a wind farm. Wu and Porté-Agel (2017) theorize that the wind farm blockage effect is the combined effect of AGWs and cumulative turbine induction. The term 'regional efficiency' of wind farms was introduced by Allaerts and Meyers (2018) to characterize the impact of atmospheric
gravity waves on wind farm performance. They claim that regional efficiency is of the same order as wake-induced wind farm efficiency losses. Allaerts et al. (2018) estimated a 4 to 6% decrease in the annual energy production of an offshore wind farm caused by the blockage effect created by AGWs. On the other hand, (Lanzilao and Meyers, 2023) and (Stipa et al., 2023) argue that AGWs can also assist wind farm wake recovery because they can create a locally favorable pressure gradient at the downstream end and behind the wind farm. However, the extent of the favorable pressure and whether or not it always improves
wind farm wake recovery is still unknown. Because of the sheer scale of AGWs and their interaction with wind farms, little if any experimental data exist to quantify their effects on wind farms; most investigations about wind farm-induced AGWs are numerical.





Numerical flow simulation, particularly large-eddy simulation (LES), is a useful tool to study AGWs. Gravity waves induced by large wind farms were first investigated with LES by Allaerts and Meyers (2017). Since then, wind farm-induced gravity waves have been studied using LES by Allaerts and Meyers (2017, 2018), Allaerts et al. (2018), and Lanzilao and Meyers (2021, 2023), all of which used pseudospectral codes and a forcing fringe region to circumvent the horizontal periodicity that is inherent in the pseudospectral approach. Stipa et al. (2023) and Maas (2022) have also studied wind farm-induced gravity wave effects using simulations with inflow and outflow boundary conditions.

The simulation of wind farm-induced atmospheric gravity waves in a finite domain requires special treatment at all domain boundaries other than the ground to stop the gravity waves from spuriously reflecting off these boundaries that do not exist in reality. AGWs propagate in all directions, so they will spuriously interact with the inflow, outflow, and top boundaries (all of which do not exist in reality). Unfortunately, current approaches to avoid wave reflection at domain boundaries are all ad-hoc and require extensive fine-tuning (Allaerts, 2016; Lanzilao and Meyers, 2023). Thus, setting up reflection-free simulations is currently tedious, computationally expensive, and time-consuming, and there is a lack of clear guidance.

The goal of this study is to make the process more systematic by investigating possible relations between the simulation setup and the physical parameters driving the flow through wind farms under stable atmospheric conditions. We anticipate that appropriate simulation parameters like domain length and height, etc, are related to internal wave properties such as wavelength and direction. We therefore investigate the proper scaling of simulation parameters and their effectiveness in avoiding wave reflections for a range of physical parameters. Understanding the relation between the physical parameters, the internal wave properties, and the simulation setup, enables recommendations to be made on how to set up simulations involving internal atmospheric gravity waves.

As this is, to our knowledge, the first systematic LES study to investigate the relation between simulation setup and physical parameters, we restrict ourselves to the study of internal gravity waves in the free atmosphere for reasons of simplicity. Investigating the free atmosphere separately is vital to handling the reflections, as IGWs propagate both horizontally and vertically and hence can reflect off all domain boundaries. To this end, we consider two linearly stratified flow scenarios: the flow over a bell-shaped 2D hill, and the flow through a wind farm canopy as a simpler surrogate for a wind farm consisting of discrete turbines. The former flow scenario can be solved analytically and is used mainly to validate the simulations and to understand the dependence of AGW properties on the governing flow parameters.

The paper is structured as follows: Section 2 gives an overview of current approaches to avoid AGW reflection at domain boundaries and highlights why it is so challenging to obtain a good simulation setup for inflow/outflow type of simulations. Next, Section 3 describes the flow scenarios studied in this study. In Section 4, the simulation methodology and the methods to analyze the LES data are explained. The numerical models and simulation setups for both flow scenarios are also explained in this section. Section 5.1 presents the results for the hill case, while Section 5.2 contains the results for the wind farm canopy. In Section 6, the conclusions of this research are presented as recommendations on how to effectively and efficiently set up simulations involving atmospheric gravity waves.



## 2 Current practices to avoid spurious AGW reflection

Two main methods exist to mitigate spurious AGW-domain boundary interaction: radiative boundary conditions and damping layers. Radiative boundary conditions often take the form

$$\frac{\partial \phi}{\partial t} + c_j \frac{\partial \phi}{\partial n_j} = 0 \tag{1}$$

where $\phi$ is the quantity being transported through a boundary over time $t$, $n_j$ is the displacement in the direction $j$ normal to the boundary, and $c_j$ is the wave transport velocity in the normal direction. One way to imagine this boundary condition is that the transported quantity $\phi$ is set to be exactly the same value as that imparted by the wave as it propagates to the boundary, thus absorbing the wave. One major difficulty is accurately determining $c_j$. This is often only possible in idealized situations. The radiative boundary condition is capable of allowing wave energy out of the domain only if the waves are moving perfectly

normal to the boundary (Durran, 1999). However, internal gravity waves are typically moving at an angle relative to horizontal.

A popular alternative to the radiative boundary condition is a damping layer. Damping layers, sometimes referred to as "sponge zones," attenuate waves as the wave moves through the zone. These zones are placed adjacent to domain boundaries and have a prescribed thickness. A downside to using such zones is that, because they have thickness, the domain size needs to be increased accordingly, which adds some extra computational cost. Typically, damping layers are either of the viscous type

or of the relaxation type, with the latter also known as Rayleigh damping layers (RDLs). RDLs are more commonly used than viscous damping and are considered more effective than the latter. For example, a recent study by Lanzilao and Meyers (2023) found that RDLs outperform radiative boundary conditions in minimizing AGW reflections in wind farm simulations. For this reason, we focus on the application of RDLs in this study.

In principle, the quantity of interest passing through an RDL, usually velocity but also sometimes temperature, is relaxed

towards a prescribed reference value with a specified time scale as the wave travels through the RDL, reflects off the boundary, and travels back through the RDL once again. Rayleigh damping is introduced into the transport equations as a forcing term which, for the case that the quantity to be damped is a three-dimensional field $\phi(\mathbf{x})$, takes the form

$$f^{RDL}(\mathbf{x}) = -\frac{1}{\tau} f(\mathbf{x}) \cdot (\phi(\mathbf{x}) - \phi_{\text{ref}}(\mathbf{x})) \tag{2}$$

where $\phi_{\text{ref}}(\mathbf{x})$ is the reference value towards which the quantity of interest in a parcel of fluid is driven along a streamline. For

example, where the velocity field is required to be damped then $\phi(\mathbf{x}) = u(\mathbf{x})$, and $\phi_{\text{ref}}(\mathbf{x}) = u_{\text{ref}}(\mathbf{x})$. In this case, $u_{\text{ref}}(\mathbf{x})$ could be defined as $[G_x, G_y, 0]$ where $G$ is the geostrophic wind vector. Sometimes the situation is ageostrophic, and a reference horizontal velocity is not appropriate, in which case only the vertical ($z$) component of velocity is damped towards zero. The Rayleigh damping function, $f(\mathbf{x})$, is a critical part of an RDL that ensures that the wave gradually dissipates energy as it travels through the layer and that the attenuation does not cause waves to reflect from the interface between the non-damped

and damped regions. The function $f(\mathbf{x})$ can be a linear, exponential, polynomial, or cosine function. A cosine function is commonly used in the vertical, which, for the upper boundary, typically has the form (Lanzilao and Meyers, 2023)

$$f(z) = \left[ 1 - \cos \left( \frac{\pi}{s_{\text{ra}}} \frac{z - (L_z - L_d)}{L_d} \right) \right] \tag{3}$$



where $z$ is height above the ground, $L_z$ is the height of the computational domain, $L_d$ is the thickness of the damping layer, and $s_{ra}$ is a constant which controls the gradient of the damping function in the vertical. This was tuned by Lanzilao and Meyers

(2023) and the best results were obtained for a value of $s_{ra}$=2. Although this suggests some dependency of wave damping on the shape of the damping function, Perić (2019) found that the form of the damping function had little impact while investigating different approaches to damping internal waves.

The time scale, $\tau$, or damping coefficient, controls how fast the quantity is relaxed toward a reference value. In Lanzilao and Meyers (2023), $\tau$ is scaled with the Brunt-Väisälä frequency ($N$) in pursuit of the optimal value for wind farm simulations,

suggesting an optimum value of $\xi = 1/(\tau N)$ of 3, with $s_{ra}$=2, as mentioned above. By contrast, Allaerts (2016), tuned the RDL damping coefficient and found an optimum value of $1/\tau = 0.0001 \ \mathrm{s}^{-1}$, translating into a typical value of $\xi = 0.017$, though it should be noted that this was for a value of $s_{ra}$=1; the same value of $\xi$ was used in a number of later studies including: Allaerts and Meyers (2017); Allaerts et al. (2018); Allaerts and Meyers (2018).

The damping layer thickness is an important parameter as it determines the space available for the forcing to dissipate the

incoming wave energy. Klemp and Lilly (1978) suggested the thickness to be more than one vertical wavelength based on hydrostatic flow solutions of a single Fourier mode. More recent studies also follow this convention but without any further investigation: Allaerts and Meyers (2017, 2018), and Lanzilao and Meyers (2021, 2023) use a 15 km thick RDL only at the top boundary as the vertical wavelengths in their simulations were always less than 15 km.

As mentioned above, many of the existing LES studies on wind farm interaction with the ABL are performed with codes

that are horizontally pseudospectral. Pseudospectral codes have the advantage of nearly exponential error convergence with increasing spatial resolution, but they come with the limitation that all lateral domain boundaries must be periodic. This poses a challenge for wind farm simulations because we wish to advect turbulent ABL flow into the domain and allow that flow along with that produced by the wind turbine wakes to exit the domain. Periodic boundaries normally would cause the wake flow exiting the domain to re-enter at the inflow. The solution to this problem is the introduction of a forcing fringe region

adjacent to the downstream boundary which forces the 'contaminated' inflow back toward the pre-computed or concurrently computed pure turbulent ABL inflow solution (Inoue et al., 2014). However, it is important to remember that pseudospectral codes with their periodic lateral boundary conditions effectively have no lateral boundaries. RDLs only have to be applied adjacent to the top boundary. Furthermore, the forcing fringe region used to force the downstream flow back toward the desired inflow state is a form of Rayleigh damping. This means that AGWs will not reflect off the lateral boundaries in pseudospectral

LES, rather they will simply exit the domain on one side and re-enter at the inflow. The forcing fringe region will have some effect on damping the lateral progression of the AGWs or may trigger spurious gravity waves, requiring an additional treatment (Lanzilao and Meyers, 2021).

As opposed to pseudospectral simulations, simulations with inflow/outflow boundary conditions, which are more commonly used in engineering applications, do not have lateral periodicity and hence do not need a forcing fringe region. Inflow is simply

injected into the domain on the inflow boundary with a Dirichlet condition and the outflow boundary is often a Neumann or advection boundary condition that lets the waked flow exit the domain. There is no need for a fringe zone that forces the waked outflow back to the desired clean inflow. On the downside, this approach requires real lateral boundary conditions that

gravity waves will either reflect off or spuriously interact with. Therefore, RDLs must be placed adjacent to these boundaries as well as adjacent to the top boundary. Very little guidance is given in the literature on how to effectively simulate gravity

waves using inflow/outflow boundaries. Stipa et al. (2023) and Maas (2022) used inflow/outflow boundary conditions for their large wind farm LES, but the approach is complicated when modelling AGWs. Any inconsistency between the inflow and the internal flow field, triggers non-physical waves at the inlet which propagate into the domain over time. Besides the generation of spurious waves due to these mismatches, avoiding gravity wave reflections from the domain boundaries is a significant challenge. Reflections are the spurious waves that are triggered by having a boundary that is not there in real but required in

these simulations. As shown in Figure 1, the gravity waves triggered by a small hill under linearly stratified conditions should move up and out of the numerical domain. But the reality of a finite domain with boundary conditions that do not exactly depict the actual physical conditions at the boundaries causes these waves to reflect. The non-dissipated wave energy accumulates at the boundaries and propagates back into the domain eventually making the solution non-physical.

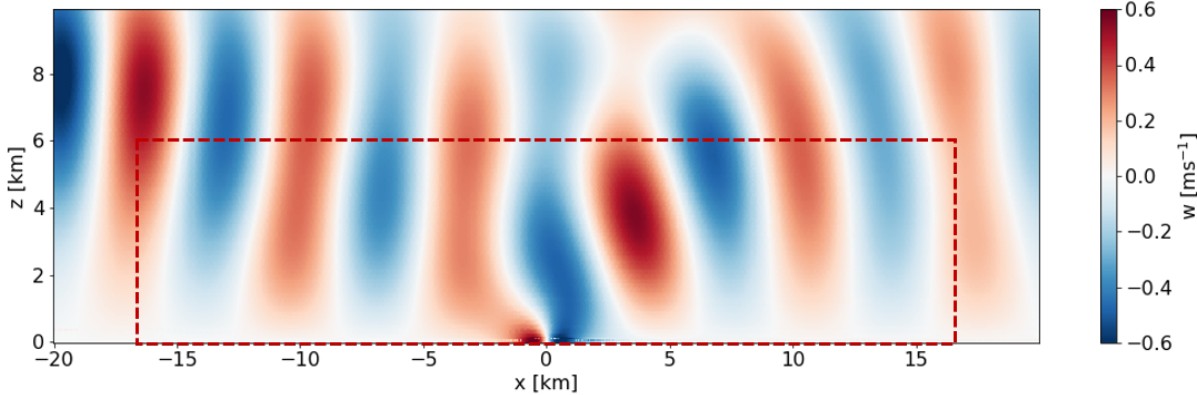

**Figure 1.** Vertical velocity contours of the flow over a small hill located at x = 0 km, produced by simulating a bell-shaped hill under linearly stratified laminar conditions in SOWFA. The region outside the red dotted box consists of Rayleigh damping layers at the inlet (left), top, and outlet (right).

The current approaches to setting up RDLs in wind farm simulations are rather ad-hoc. Extensive fine-tuning of the damp-

ing parameters is required to set up a working simulation which may be applicable only to a specific case (Allaerts, 2016; Lanzilao and Meyers, 2023). As a result, setting up reflection-free simulations is tedious, computationally expensive, and time-consuming. The aim of this paper is to investigate how to make the process less ad-hoc by investigating relationships between the simulation setup and the fundamental physical parameters driving wind farm flow under stable atmospheric conditions.

## 3    Flow Scenarios

In this study, we consider two simple, two-dimensional flow scenarios that generate internal gravity waves: the flow over a bell-shaped hill, and the flow through a wind farm canopy. Our initial focus lies on the hill case, which is used in various





meteorology and earth sciences studies, such as Vosper and Ross (2020) and Snyder et al. (1985). The reasons to start with the hill case include the simplicity of the flow scenario, computational affordability, and the availability of a semi-analytical solution, which is useful for validation purposes. Moreover, the primary aim of this study is how best to handle atmospheric gravity waves in numerical simulations, and the wave source is thereby of less importance. Nevertheless, the simulated hill heights are similar to typical wind turbine rotor tip heights, and the half-width of the hill is of the same order as typical wind farm lengths. In the second flow case, we study the flow through wind farm canopies to extend the findings from the hill case to wind farm applications. Unlike the rigid hill, a wind farm canopy is a porous region in which the drag force of the wind turbines is applied homogeneously. The details of both flow scenarios are discussed in more detail in the following subsections.

## 3.1 Scenario 1: Bell-shaped hill

The first flow scenario considers flow over a 2D bell-shaped hill. The height profile $h$ of a bell-shaped hill – sometimes referred to as the Witch of Agnesi (WOA) profile – is defined in the horizontal direction $x$ as

$$h(x) = \frac{H}{1 + (x/L)^2}. \tag{4}$$

This profile is governed by two parameters, i.e., the maximum hill height $H$ and the half-width at half-height $L$.

To allow a direct comparison between an LES and a linear gravity wave potential flow solution, surface friction and the Coriolis force were excluded, and only a linearly stratified free atmosphere was considered with uniform inflow. Although the inflow was laminar, the LES model could still generate turbulence due to flow separation behind the hill.

A computationally inexpensive semi-analytical flow solution over the hill was used to understand the properties of the IGWs and validate the numerical solutions. We use the analytical solution along with the R-RMSE metric (discussed in Section 4.3) to quantify reflections in the simulations. The following subsection gives an overview of this semi-analytical solution and how it is used in this study.

### 3.1.1 Semi-Analytical Solution

Linear wave theory, particularly the Taylor-Goldstein equations, is commonly used to study atmospheric gravity waves. Allaerts (2022) used these equations to develop a Python module called Linear Buoyancy Wave Package (LBoW) to solve linear buoyancy wave problems. In this research, we use a part of this code that computes a semi-analytical steady-state solution of the uniform, stratified flow over the WOA hill. The code solves the equations on a grid in frequency space using a Fast Fourier Transform (FFT). The solution is independent of the grid size in the vertical direction but not in the horizontal. A solution at any vertical level can be acquired without requiring a prior solution at lower and higher levels. The grid resolution in the horizontal direction dictates the solution accuracy. The FFT solution deviates from a theoretical Fourier transform of the bell-shaped hill for high horizontal wave numbers due to rounding errors. Mesh size in the range of $20\,m$ to $100\,m$ is recommended to compute the semi-analytical solution.

The steady-state solution of uniform flow over the hill in a stably stratified free atmosphere with a uniformly increasing potential temperature with height is shown in Fig. 2 , where the upward flow deflection by the hill triggers only IGWs. The





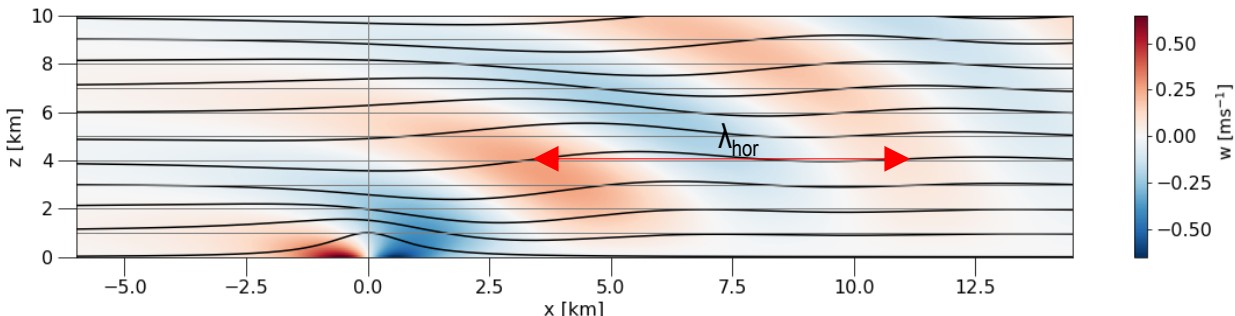

**Figure 2.** Steady-state analytical solution of flow over a WOA hill centered at $x = 0km$. The color contours represent vertical velocity and the streamlines show the propagation of the waves. Allaerts (2022)

vertical velocity contour and streamlines clearly show the propagation of gravity waves in the vertical direction. Depending on flow conditions, the waves can be propagating or evanescent; the energy dissipates with height for the latter. It is important to understand that the wave train seen in Fig. 2 is not mono-chromatic but is actually a wave spectrum, where the wave properties depend mainly on the free-atmosphere Froude number and the hill shape and size. There are three important wavelengths to consider:

1. The wavelength with the highest amplitude is the *dominant* wavelength, which depends on the hill width.

2. However, the dominant wavelength is not necessarily the wavelength that is most visually apparent in contour plots. For example, that most apparent wavelength, which is termed the *effective* wavelength, is shown in Figure 2 with red arrows and labeled $\lambda_{hor}$ .

3. Lastly, there is a *critical* wavelength (often we consider its reciprocal, the *critical* wavenumber). For wavelengths smaller (or wavenumbers larger) than the critical one, the waves cannot be supported by buoyancy so they dissipate and are called "evanescent" waves.

The properties of IGWs depend on the wind speed $U$, hill half-width at half-height $L$, hill maximum height $H$ and Brunt-Väisälä frequency $N = \sqrt{(g/\theta)(\frac{\mathrm{d}\theta}{\mathrm{d}z})}$, where $g$ is acceleration due to gravity and $\theta$ is the potential temperature at height $z$. Employing similarity theory, we normalize these variables to get two physical parameters, the Froude Number (Fr) and the Slope Parameter ($S_h$), where $Fr = U/NL$ and $S_h = H/L$.

Although the analytical solution tells us that wave amplitude depends more on the hill slope, particularly the height, the effective wavelengths and the orientation of the IGWs depend on $Fr$. The effective horizontal wavelength, ($\lambda_{hor}$) is sensitive to the hill width for low $Fr$, but for $Fr > 0.5$, it depends more on the Scorer parameter ($N/U$), which is the reciprocal of the buoyancy length. This can be seen in Fig. 3a, where the normalized $\lambda_{hor}$, calculated from the semi-analytical solution and LES, is plotted against $Fr$. We measure $\lambda_{hor}$ as twice the distance between the global maxima and minima determined from a vertical velocity plot at a constant height along the domain length. We see that as $Fr$ is increased, $\lambda_{hor}$ increases, but it is not a linear relation.





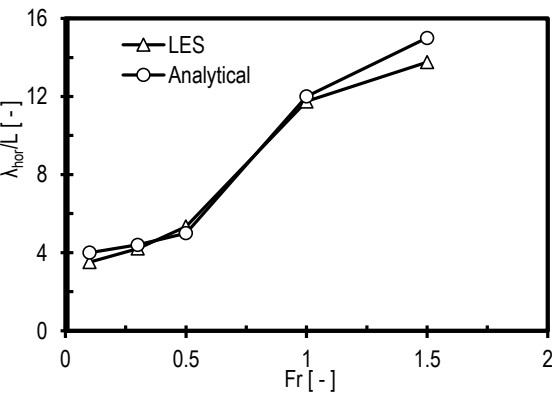
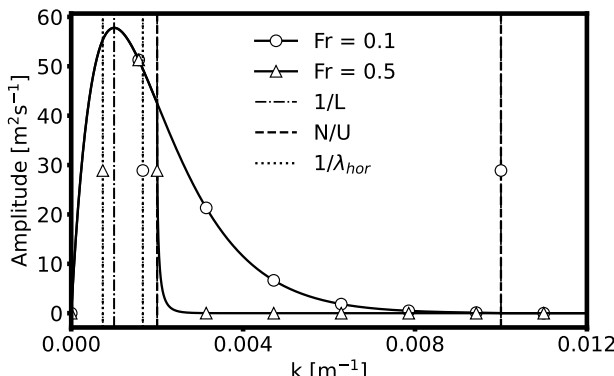

(a) The variation of effective horizontal wavelength with $Fr$     (b) The streamwise wave spectrum for vertical velocity at a height

**Figure 3.** (a) Effective horizontal wavelength as a function of $Fr$, and (b) effective, dominant and buoyancy wavelengths identified for $Fr = 0.1$ and $0.5$.

Although we visually identify with the effective wavelength of AGWs when we view figures like Figure 2, those AGWs are really a spectrum of waves, as shown in Figure 3. On this spectrum, we also show the three important length scales. The dominant wavenumber is $1/L$. The situation can be classified into two different conditions:


1. When the critical wavenumber is greater than $1/L$ or when $Fr < 0.5$, the dominant wavelength is greater than $\lambda_{hor}$. In this situation, $\lambda_{hor}$ depends more on the hill half-width then the Scorer parameter, and the entire wave spectrum is preserved and the waves propagate.

2. On the other hand, when $Fr > 0.5$, the critical wave number is closer to $1/L$; thus, a part of the spectrum becomes evanescent, and wave numbers greater than $N/U$ are dissipated. In this case, the dominant wavelength is less than $\lambda_{hor}$,

which depends more on the Scorer parameter.

The effective vertical wavelength ($\lambda_{ver}$) depends only on the Scorer parameter, which is proportional to the vertical wave number. It can be seen that the effective horizontal and vertical wavelengths can be greater than the dominant length scales, i.e., the hill width and height; thus, domains scaled with the hill width and height might be inappropriate for the accurate simulation of gravity waves.

This study focuses only on the Froude number as it concerns gravity wave properties that are critical to the simulation setup. The slope parameter is less critical for this research and the slope of the hill ($S_h$) is kept the same. With different hill shapes, the amplitude and wave spectrum would change, making comparisons between simulations with different hill shapes inconsistent. In addition, we only consider the steady-state solution.





### 3.2 Scenario 2: Wind Farm Canopy

Wind farm flow interactions with the atmosphere involve a wide range of length scales. When focusing on wind farm-induced IGWs, large length scales are important as the expected IGW wavelengths are on the scale of the wind farm length which can be several kilometers. Since the intra-farm (turbine wake) interactions is not the focus of this study, wind farm canopies are a convenient way to model wind farms without the complexity of modelling individual wind turbines. The concept of a wind farm canopy was introduced by Markfort et al. (2018) through an analytical model to represent large wind farms in weather 255 prediction models. In our work, we use a similar approach to simulate the cumulative drag force of a wind farm consisting of a number of wind turbines of a given type.

The standard coefficient of thrust associated with a wind turbine rotor is

$$C_t = \frac{T}{\frac{1}{2}\rho U^2 A_r} = \frac{T}{\frac{1}{2}\rho U^2 \pi \frac{D_r^2}{4}}, \tag{5}$$

where $A_r$ is the rotor swept area, $D_r$ is turbine rotor diameter, $\rho$ is the density of air, $T$ is dimensional rotor thrust force, and $U$ 260 is the freestream velocity. Because we are distributing the force of the rotors in the wind farm over the volume occupied by the wind farm, it is better to define a thrust coefficient based on the area of the farm belonging to this rotor (i.e., the entire footprint area considering turbine spacing alotted to this wind turbine). This area is $A_f = (S_x D_r)(S_y D_r) = S_x S_y D_r^2$ where $S_x$ and $S_y$ are the turbine spacings in the two horizontal directions within the farm. This new thrust coefficient is

$$c_t = \frac{T}{\frac{1}{2}\rho U^2 S_x S_y D_r^2}. \tag{6}$$

The two thrust coefficients are related to each other through

$$c_t = \frac{\pi}{4 S_x S_y} C_t. \tag{7}$$

Defining dimensional thrust based on $c_t$, we have

$$T = \frac{1}{2}\rho U^2 S_x S_y D_r^2 c_t, \tag{8}$$

and normalizing dimensional thrust by volume of the farm occupied by a turbine, $V_f = S_x S_y D_r^2 (H_t - H_b)$, where $H_t$ and $H_b$ 270 are the height of the top and bottom of the turbine rotor and also the height of the top and bottom of the wind-farm canopy, we have

$$f_{wf} = \frac{T}{V_f} = \frac{1}{2}\frac{\rho U^2 S_x S_y D_r^2}{S_x S_y D_r^2 (H_t - H_b)} c_t = \frac{1}{2}\frac{\rho U^2}{(H_t - H_b)} c_t = \frac{1}{8}\frac{\rho U^2 \pi}{S_x S_y (H_t - H_b)} C_t. \tag{9}$$

It is this thrust per unit volume that is applied to the momentum equations as a source term within the wind turbine canopy volume. The wind speed, $U$, is determined locally at each cell centers within the canopy.





**Table 1.** Key variables in the simulation of atmospheric gravity waves under linearly stratified free atmospheric conditions with typical ranges in wind energy applications.

| Variable | Variable Type | Range |
|---|---|---|
| Velocity ($U$) | Flow | 1 ms$^{-1}$–25 ms$^{-1}$ |
| Brunt-Väisälä frequency ($N$) | Flow | 0.005 s$^{-1}$–0.02 s$^{-1}$ |
| Half-Width of Hill or Canopy Length ($L$) | Shape | 1 km–15 km |
| Hill or Canopy Height ($H$) | Shape | 16 m–240 m |
| Domain Length (undamped) ($X$) | Simulation | 0.5 km–200 km |
| Domain Height (undamped) ($L_z$) | Simulation | 0.3 km–40 km |
| Damping thickness ($L_d$) | Damping characteristic | 0.3 km–45 km |
| Damping coefficient ($1/\tau$) | Damping characteristic | 0.001 s$^{-1}$–0.5 s$^{-1}$ |

## 4 Methods

### 4.1 Simulation Parameters and Setup

A set of non-dimensional parameters govern the flow over terrain and through wind farms under linearly stratified atmospheric conditions. These parameters can be determined by normalizing the flow equations or performing dimensional analysis of a number of key variables. These variables are detailed in Table 1 along with their practical values in wind energy applications. From these variables, the set of non-dimensional parameters in Table 2 can be defined. The first two are physical parameters, namely the Froude number and slope parameter, whilst the remainder are simulation parameters.

Appropriate choice of grid structure, resolution, and time step are important in the simulation of gravity waves, particularly for very low values of $Fr$. Because the flow interactions can trigger subgrid-scale wavelengths. Likewise, the frequencies of some waves in the spectrum could be shorter than the simulation time steps, leading to an unresolved fraction of the spectrum. However, the relevant value of $Fr$ for wind farm applications is approximately between 0.1 and 0.5 for which a grid independence study was carried out. It was found that grid resolutions, roughly 10 m in all directions, used in wind farm LES to resolve the wind turbines and their wakes, is more than sufficient to resolve the wind-farm-induced AGWs.

LES of flow over the WOA hill and through the wind farm canopy is carried out with Simulator for Wind Farm Applications (SOWFA) Churchfield et al. (2012a). Based on OpenFOAM, this code is mainly used for the LES of atmospheric flows over terrain and through wind farms, where a one-equation model is commonly used for sub-grid-scale turbulence modelling. SOWFA has actuator models for wind turbine aerodynamics that can be coupled with aero/servo/elastic tools. Moreover, it can use boundary inflow data from mesoscale weather data, and terrain can be included through non-conformal meshes. The model set-up used solves the incompressible Navier-Stokes equations under non-hydrostatic conditions and with the Boussinesq approximation for buoyancy. Equations for continuity, momentum, and potential temperature are those typically used in the LES of atmospheric flows. A more complete description is given in Churchfield et al. (2012b). For simulations with the wind





**Table 2.** Non-dimensional parameters derived from the variables in Table 1 with typical ranges.

| Non-dimensional parameter | Definition | Range |
|---|---|---|
| $Fr$ | $U/NL$ | 0.1–0.5 |
| $S_h$ | $H/L$ | 0.016–0.4 |
| $\tilde{X}$ | $X/\lambda_{hor}$ | 0.5–6.0 |
| $\tilde{L}_z$ | $L_z/\lambda_{ver}$ | 0.3–2.0 |
| $\tilde{L}_d$ | $L_d/\lambda_{ver}$ | 0.5–2.0 |
| $\xi$ | $1/(\tau N)$ | 1–50 |

farm canopy, the drag force of the wind farm is added to the momentum equation as a body force. Rayleigh damping is applied as a body force in the forcing zones through the momentum equation, details of which are given in the following subsection.

Figure 4 shows the numerical setup for the hill and wind farm canopy cases. The flow is driven in and out of the domain by inflow/outflow boundary conditions. Periodic boundary conditions are used in the transverse direction only. Because SOWFA's

inflow and outflow boundary conditions and lower and upper impenetrable boundaries set the velocity flux locally over each boundary face, there is no need for pressure boundary conditions (keeping in mind that the pressure solve in an incompressible code enforces continuity). To simplify the set-up, wind shear is neglected and a uniform inflow is imposed as the inlet boundary condition for wind speed. In addition, free-slip boundary conditions are imposed at the top and bottom of the domain. The temperature profile is linear in the vertical direction, giving a constant Brunt-Väisälä frequency with height. There is no heat

flux at the ground. Also Coriolis forces are not considered. These conditions are intended to mimic those of a stable free atmosphere without the ABL and inversion layer.

A surface profile for the WOA hill is created using a Python script and the computational mesh conforms to the hill. A mesh with layered refinement is used with $20\ m \times 20\ m \times 20\ m$ resolution in the non-damped domain, and $40\ m \times 20\ m \times 40\ m$ in the top damping layer. The first mesh layer near the surface ends in the top damping layer to ensure any numerical noise

for switching to a coarser mesh is damped. The domain for all cases is $100\ m$ in the $y$ direction, whereas the $x$ and $z$ extents are varied as a function of the effective horizontal and vertical wavelengths, respectively, as the domain length and height are critical simulation variables when simulating gravity waves. The exact domain length and height used for each simulation are reported while discussing the results. Both the hill and wind farm canopy are extended in the transverse direction to the sides of the domain effectively creating an infinite ridge and a semi-infinite wind farm, respectively, as we are primarily interested

in the vertical and streamwise flow.

The guidelines to systematically model AGWs can be established based on the hill-scenario, however, the characteristics of the AGWs may vary for a porous wind farm canopy as opposed to a solid hill. Therefore, wind farm canopies are simulated to extend the findings from the hill-scenario to the wind farms. This approach is used to reduce computational resources, which is desirable as hundreds of cases are run to evaluate the extent of wave reflections. The wind farm canopy model can be used

with relatively coarse grids compared to conventional actuator models that resolve the wind turbines to a given extent. The





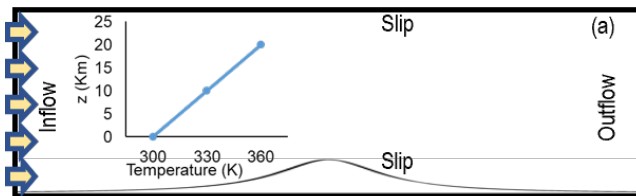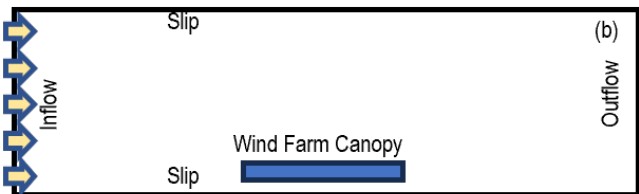

**Figure 4.** Lateral view of the simulation setup: (a) with a schematic WOA hill profile and (b) with a schematic wind farm canopy.

numerical setup with the wind farm canopy is the same as that of the WOA hill, except that the hill is replaced with wind farm drag as a body force.

### 4.2 Rayleigh Damping

RDLs are implemented as zones adjacent to the reflective boundaries in the simulation domain and far from the region of
interest, such as a wind farm. The primary role of an RDL is to dissipate the energy propagated through the zone by AGWs. In this study, only the vertical velocity is damped unless mentioned otherwise as the prominent perturbation is the vertically deflected velocity by the hill or the wind farm canopy. If periodic streamwise boundary conditions are used, an RDL is implemented at the top boundary only with an RDL-like fringe region at the outlet where flow is recycled to the inlet. However, in a simulation setup with inflow/outflow boundary conditions, RDLs may be required at various boundaries. The RDL at the inlet
will filter any incoming turbulence. Generally, damping incoming turbulence into the boundary layer is undesirable in wind farm simulations; however, in this study, for simplicity, we have neglected inflow turbulence. This should not impact the aim of the study which is to minimise gravity waves reflections in an efficient manner.

### 4.3 Quantifying Reflections

Reflectivity is quantified by the method proposed by Allaerts and Meyers (2017), which is a modification of the procedure
initially given by Taylor and Sarkar (2007). The reflection coefficient ($Cr$) is one of the two primary tools used here to analyze the simulation data, and its calculation can be summarised in the following two steps. First, the vertical velocity values on a vertical streamwise plane are converted to horizontal and vertical wave number space by implementing Fourier transforms. The upward and downward moving waves are separated based on the quadrants they fall into on the wave number coordinates.The upwards propagating waves are in the first and third quadrants, whereas the downwards propagating are in the second and fourth
quadrants. Then, the energy associated with these decomposed waves is calculated, and the reflection coefficient is estimated by taking the ratio of total downwards to upwards propagating energy. This $Cr$ metric sometimes gives inconsistent values, especially for low $Fr$ and $L$, possibly because the spectrum includes a large fraction of high-frequency waves. Therefore, visual inspection of the simulation fields, especially the vertical velocity, is critical to ensure the value predicted is realistic; improving or replacing this metric is future work.



In the case of the hill, the numerical solution can be compared to the semi-analytical solution using the Relative-Root Mean Square Error (R-RMSE) metric. The difference between the vertical velocity ($w$) fields from the numerical and analytical solutions are normalized with the maximum $w$ from the analytical solution, to determine the R-RMSE over a vertical plane in the streamwise direction. The vertical velocity is always taken at time $t$ when $t = 300/N$, which is equivalent to 2.083, 4.167, or 8.33 $hrs$ for $N = 0.02$, $0.01$ or $0.005$ $s^{-1}$, respectively. As the R-RMSE metric depends on the number of points

sampled, we only calculated this for the same section of the domain, around the hill or canopy. This has the drawback of not capturing reflections close to the boundaries. However, the R-RMSE can be better than $Cr$ in capturing localized spurious wave sources, like numerical noise at the interface of the damping layers with the non-damped domain. Due to the strengths and weaknesses of the two metrics, we analyzed the results of both, and generally, it is found that R-RMSE metric detects the same patterns seen in $Cr$. Therefore, only results in $Cr$ will be reported in the paper to be consistent with the literature. In all

cases, unless otherwise mentioned, the vertical velocity values are taken from a vertical plane at the mid-point of the domain. For reflection-free simulations, we consider the criterion $Cr$ and R-RMSE $< 10\%$ should be met.

## 4.4  Simulation Sets

Simulations were run initially to acquire a base case and then use it to explore reflection dependency on the non-dimensional parameters defined in Table 2. Various combinations of damping layers with different combinations of characteristics were

simulated to investigate the most appropriate configuration with minimum reflections. The details on the configurations explored are given in the Appendix. Besides exploring the configuration of the damping layers, three simulation sets are defined as detailed in Table 3. These three simulation sets were repeated for the wind farm canopy.

- The first set of simulations investigates the impact of damping characteristics on reflections and their optimal values for a range of $Fr$. Set 1(a) explores the extent of reflections when varying the damping coefficient for a range of $Fr$ values,

i.e., $0.1$ to $1.5$, where $Fr$ was adjusted by changing $U$ and $L$. In Set 1(b), the damping coefficient and thickness were changed when setting $Fr = 0.1$ and $0.5$, where $Fr$ was adjusted by changing either $N$ and $U$, or $U$, $N$, and $L$ ensuring dynamically similar solutions.

- Set 2 explores the impact of the domain length on wave reflections. In this case, the domain length was varied for two values of $Fr$. Hills of two half-widths, 5 km and 10 km, were simulated for each value of $Fr$.

- Set 3 investigates the impact of domain height on wave reflections. In this case, for two values of $Fr$, the domain height was varied in proportion to the expected ($\lambda_{\text{ver}}$).



**Table 3.** Simulation sets.

| Set | Parameters Investigated | | Variables Changed | | Number of Simulations |
|---|---|---|---|---|---|
| | Physical | Simulation | Physical | Simulation | |
| 1(a) | Fr [0.1, 0.3, 0.5, 1.0, 1.5] | $\xi, \tilde{L}_d$ | U, L | $1/\tau$ | 25 |
| 1(b) | Fr [0.1, 0.5] | $\xi, \tilde{L}_d$ | U, N, L | $1/\tau, L_d$ | 120 |
| 2 | Fr [0.1, 0.5] | $\tilde{X}$ | Fr, L [5, 10 km] | $X$ | 20 |
| 3 | Fr [0.1, 0.5] | $\tilde{L}_z$ | Fr | $L_z$ | 12 |

# 5 Results

## 5.1 Hill

This section first considers the optimum damping configuration for the hill case as a baseline before moving on to the wind
farm canopy set-up.

### 5.1.1 Configuring the damping layers

Correctly setting up the size and location of the damping layers is important for the accurate and effective simulation of AGWs.
Therefore, we test various configurations of the damping layers that give minimum reflections and use this as a base case for
further investigation of the sensitivity to the non-dimensional parameters. As mentioned earlier, we require that both $Cr$ and
R-RMSE are less than $10\%$ for the simulation to qualify as what we define as "reflection-free." Following a brief investigation
of various combinations of damping layers, it was found that RDLs at the inlet, outlet, and top boundaries give minimum
reflections and thus all subsequent simulations use this configuration as the base case.

### 5.1.2 Dynamic Similarity

We started the analysis by testing whether the non-dimensional parameters discussed in Section 4.1 are sufficient to ensure
dynamic similarity, i.e. the non-dimensional solution remains the same if the non-dimensional parameters are the same irre-
spective of the values of the variables defining them. The simulations performed had a setup with a damping layer thickness
and domain height $1.5$ times the effective vertical wavelength and a domain length five times the expected effective horizon-
tal wavelength. Fig. 5 shows the dependency of wave reflections on the non-dimensional damping parameter, $\xi$ for changing
buoyancy and hill half-widths ($U/N$ and $L$), and buoyancy and advection timescales ($1/N$ and $L/U$). In terms of minimising
reflections, the plots indicate that a suitable range for $\xi$ is between 1 to 10 when $Fr = 0.1$, and the buoyancy time scale is
an appropriate scaling parameter for the damping coefficient. The plots further show that the results are independent of both
the time and the length scales, because the solutions are dynamically similar. The buoyancy length was kept constant (i.e.
$U/N = 0.5$ km) by changing the Brunt-Väisälä frequency and inflow velocity for a constant hill half-width (i.e. $L = 5$ km).





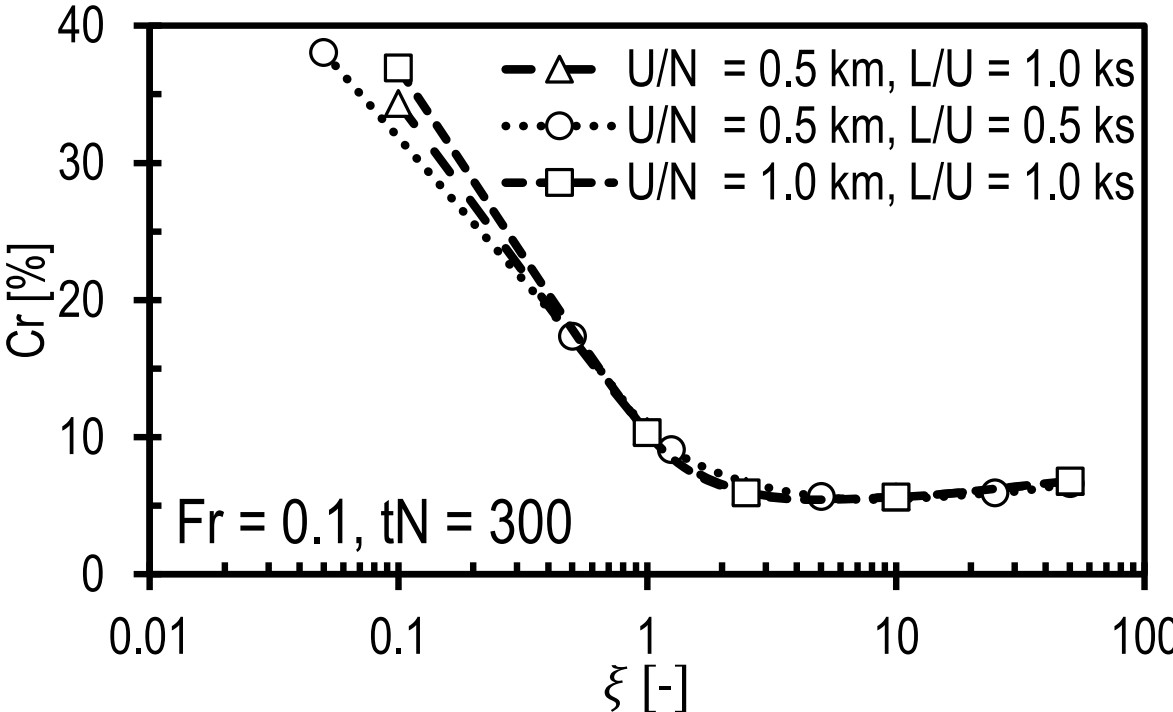

**Figure 5.** Reflection coefficient as a function of normalised damping coefficient $\xi$ for varying length and time scales.

Thus, the advection time scale was different for these two cases. Whereas, the buoyancy length was varied by adjusting the
Brunt-Väisälä frequency and hill half-width to fix $Fr = 0.1$; the buoyancy time and physical length scales were varied. These
three cases can be compared in different ways and it will establish dynamically similar solutions for all comparisons. Similar
results are seen for $Fr = 0.5$, though these are not shown for clarity.

### 5.1.3  Optimal Damping Coefficient as a function of Froude number

In general, very low damping coefficients lead to the highest reflections; however, very high damping coefficients, e.g. $\xi >> 10$,
also enhance reflections as the IGWs reflect off the damping layer instead of the boundaries. These effects can be seem in Fig. 6.
Strong reflections and energy accumulation can be seen visually in Fig. 6 (top) for $\xi = 0.1$, where they appear parallel to the
inlet on the left and gradually contaminate the solution in the entire domain. Figure 6 (middle) shows vertical velocity contours
when $\xi = 10$ that are least affected by reflections and energy accumulation at the inlet. Moreover, the gradual decay of IGWs
with height inside the top RDL is evidence of suitable damping characteristics. Whereas, Fig. 6 (bottom) shows vertical velocity
contours when $\xi = 50$, where IGWs are abruptly attenuated right at the start of the top damping layer. Also accumulated waves
appear at the end of inflow RDL as if it were a hard boundary.



It was shown in Section 5.1.2 that the optimal damping coefficient for $Fr = 0.1$ is around 10. However, Fig. 7 shows that for values of $Fr > 0.1$, the optimal damping coefficient is somewhat less with minimum reflections occurring for values of $\xi$ between 2 and 10, although the sensitivity to $\xi$ in this range is small. The adverse level of reflection for low and very high damping coefficients is also apparent from this plot. For $Fr = 0.1$ and 1.5, $Cr$ cannot be made less than $10\%$, because the damping layer thickness is just one vertical wavelength which is insufficient for these very low and supercritical values of $Fr$, respectively. These simulations had damping layer thicknesses and domain heights equal to the effective vertical wavelength, and the domain length was twice the effective horizontal wavelength. Supercritical values of $Fr$ ($Fr > 1$), a regime in which much of the wave content is evanescent, are less likely to occur for large wind farms, whereas low $Fr$ values are very likely. For this reason, the following sections will restrict analysis to the expected upper and lower ranges of the Froude number for wind energy applications, i.e. $Fr = 0.1$ and 0.5.

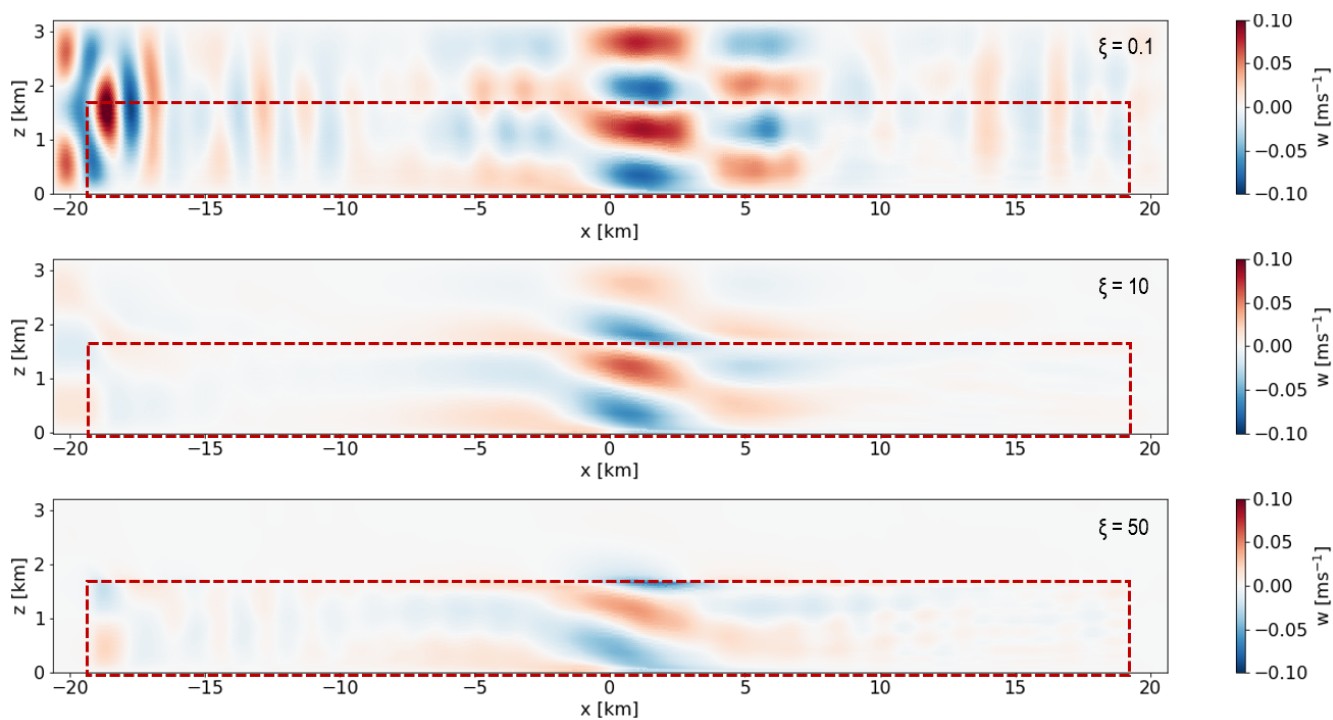

**Figure 6.** Contours of vertical velocity in a streamwise-oriented vertical plane at $tN$ 300 for $Fr = 0.1$ with $\xi = 0.1$ (top), $\xi = 10$ (middle), and $\xi = 50$ (bottom). The red box shows the non-damped domain, and everything outside is the RDL.

### 5.1.4   Impact of Damping Layer Thickness on Reflections

So far, we have considered the impact of $\xi$ on the reflections while $L_d$ is equal to or greater than the vertical wavelength. However, the impact of damping characteristics on reflections is coupled. A weak, thick damping layer may have the same impact as a strong, thin layer. Therefore, determining the coupled impact of the damping characteristics on reflections and

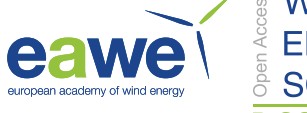
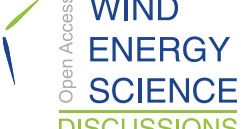


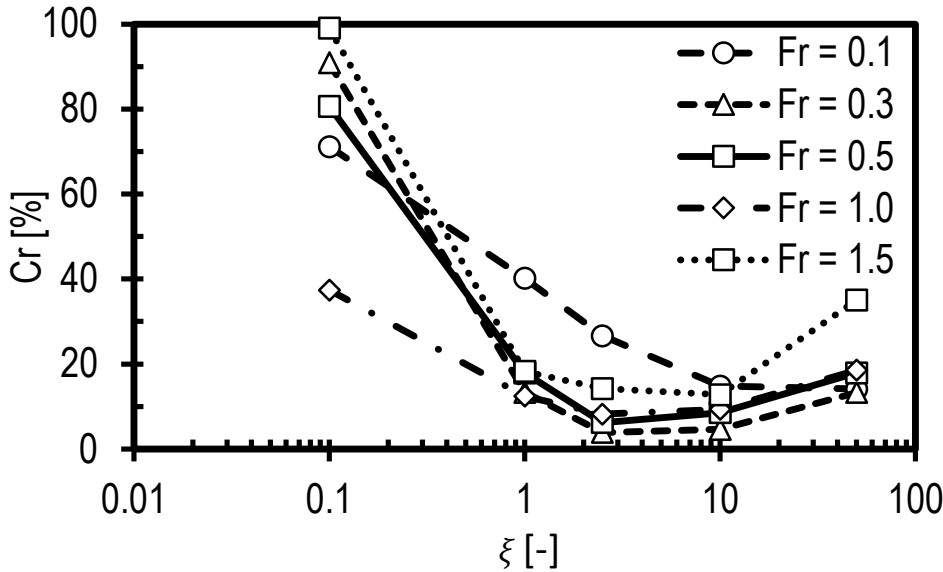

**Figure 7.** Reflection coefficient as a function of $\xi$ for a range of Froude numbers.

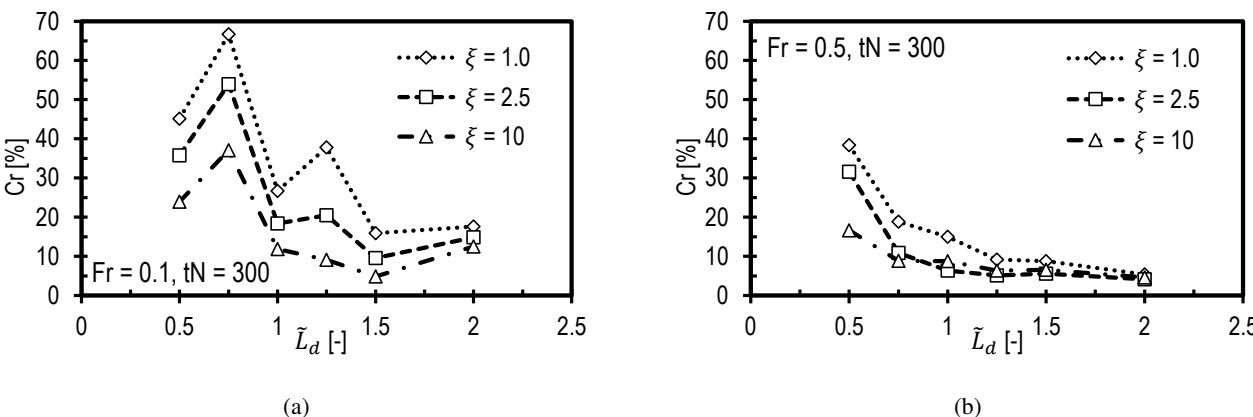

| (a) | (b) |

**Figure 8.** Reflection coefficient for the hill case as a function of normalised damping layer thickness for different values of the damping coefficient when (a) $Fr = 0.1$ and (b) $Fr = 0.5$.

the minimum damping thickness is desirable, as knowing the minimum effective damping layer thickness can help reduce the computational load.

The joint effect of different damping parameters is investigated for $Fr = 0.1$ and $Fr = 0.5$, where the damping thickness and coefficient are varied simultaneously. In general, thicker damping layers for all damping coefficient values should reduce reflections but thinner damping layers are desirable for low computational cost. Figure 8 shows that a damping layer set to the optimal damping coefficient and accommodating one effective vertical wavelength seems more effective than a thicker layer





with a sub-optimal value of $\xi$. Figure 8a shows that $Cr$ is minimum for all values of $\tilde{L}_d$ when $\xi$ is 10. The non-monotonic nature of the plots in this figure indicate how optimizing the setup for a low value of $Fr$ is more challenging. The exact reason for this non-monotonic behavior is hard to establish but it is most likely associated with the complicated wave properties for low values

of $Fr$. For instance, wavelengths become shorter for low $Fr$. Thus, the resolved wave spectrum can vary significantly for even small changes in the domain size, which is linked with the amount of wave reflection. Furthermore, the waves are more aligned to the horizontal for a low value of $Fr$, complicating the interaction with the background advecting flow as the wave fronts are more directly aligned with the inlet wind flow than that of a higher $Fr$ case. Moreover, the energy accumulation is higher for a low value of $Fr$ as the wave speed is faster than the advection speed in turn causing more contamination of the solution.

This also hints at inadequacy of the RDLs in terms of their ability to prevent energy accumulation at the inlet only delaying its propagation back into the domain.

Figure 8b shows monotonic reflection behavior for $Fr = 0.5$. It can be seen that a value of $\xi = 2.5$ provides an optimal solution when $\tilde{L}_d > 1.0$, and a value of $\xi = 10$ is best when $\tilde{L}_d < 1.0$. The damping layer thickness required when $Fr = 0.1$ is slightly bigger than that for $Fr = 0.5$ to limit the reflections to the same levels. This can be seen when comparing the optimal

setups for $Fr = 0.1$ and $Fr = 0.5$, where $Cr$ is limited to $5\%$ whereby $\tilde{L}_d$ is 1.2 for $Fr = 0.5$ and $\tilde{L}_d$ is 1.5 for $Fr = 0.1$.

In summary, the damping layer thickness should be greater than one effective vertical wavelength to efficiently damp the IGWs. This aligns with the recommendation of (Klemp and Lilly, 1978) but is based on analyzing the entire wave spectrum instead of analyzing individual wave numbers, which was their approach.

### 5.1.5   Domain Length Impact on Reflections

Intuitively, the location of the top boundary is more important than that of the other boundaries, as the gravity waves travel upwards, and reflections are mainly expected from the top boundary. However, the accumulation of energy at the inlet boundary, shown in Fig. 1 and Fig. 6, indicates the importance of appropriately positioning the inlet and outlet relative to the hill or wind farm (i.e. the zone of interest). One approach is to place boundaries far from the zone of interest to prevent spurious waves from affecting the flow around the zone of interest. This approach has two major flaws: it is costly in terms of computational resource

for LES studies and simulations can run only for a limited time before the reflections reach the zone of interest. This constraint conflicts with the common practice in wind farm LES of simulating several domain flow-through times to obtain reliable statistics. Moreover, the contamination of the solution upstream of a wind farm may affect the inflow, and an unrealistic inflow to the wind farm would lead to an unreliable solution. Therefore, knowing the shortest possible domain length that ensures the least contaminated inflow approaching the zone of interest is important to produce accurate solutions and save computational

resources and time. To this end, a set of simulations, denoted in Table 3 as Set 2, were performed with two hill half-widths at half-height for values of $Fr = 0.1$ and $Fr = 0.5$, where the domain length was varied between $0.5\lambda_{hor}$ and $4.0\lambda_{hor}$. Instead of simulating for several different values of the damping parameters, we used the results from Section 5.1.4 with $\xi = 10$ for $Fr = 0.1$ and $\xi = 2.5$ for $Fr = 0.5$. The value of $\tilde{L}_d$ was set to 1.5 for $Fr = 0.1$ and 1.2 for $Fr = 0.5$.

Figure 9a shows the impact of domain length on reflections. The reflection coefficient shows the same trend for $Fr = 0.1$ and

$Fr = 0.5$ with $L = 5$ and $10\ km$. The domain length should be at least one effective horizontal wavelength, as $Cr$ increases





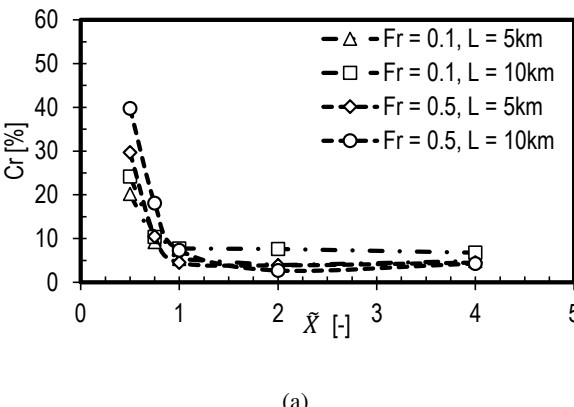
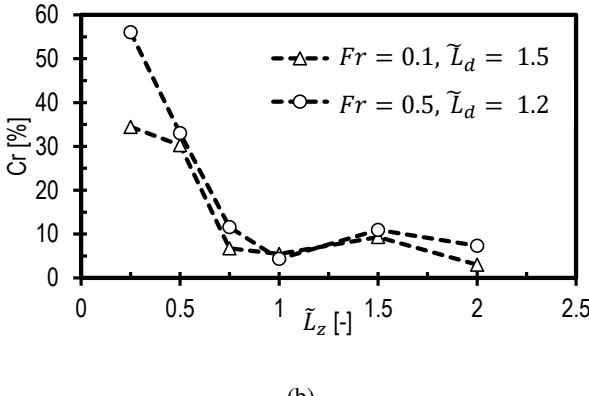

|     (a)     |     (b)     |

**Figure 9.** Reflection coefficient for the hill case as a function of: (a) domain length for $Fr = 0.1$ and $Fr = 0.5$ for $L = 5$ km and 10 km; and (b) domain height for $Fr = 0.1$, $\tilde{L}_d = 1.5$ and $Fr = 0.5$, $\tilde{L}_d = 1.2$.

abruptly for shorter domains. We emphasize the discussion in Section 3.1 concerning the variation in effective horizontal wavelength with $Fr$. It was established that $\lambda_{hor}$ depends on both $L$ and $U/N$ for $Fr < 0.5$ but predominantly on $U/N$ for $Fr > 0.5$. In other words, the domain length should not be scaled with $L$. Instead, the expected effective horizontal wavelength should be calculated from linear theory and this value used to set the domain length to accommodate at least one effective

horizontal wavelength. It can be seen that domain lengths over $\lambda_{hor}$ are unnecessary as $Cr$ barely decreases for increasing domain length. The evolution of reflections in time is another critical aspect to highlight, as we observed increasing $Cr$ values in time in all simulations. We run the simulations to a steady-state, which in general takes only a few flow-through times and domain length equal to $\lambda_{hor}$ is sufficient for these short runs. For longer runs, like diurnal cycles, the wave energy accumulation at the inlet may eventually contaminate the solution, which is a topic for future. It is important also to note that the simulated

domains are symmetric about the hill top, i.e. the distance between the inlet and outlet is the same from the center. In wind farm simulations, there should be a minimum distance between the wind farm and the inlet to allow the flow to adjust to the pressure field created by the AGWs avoiding non-physical blockage (Lanzilao and Meyers, 2023). This minimum requirement will be the subject of a further study.

### 5.1.6    Impact of Domain Height on Reflections

It is expected that the height of the domain should scale to the effective vertical wavelength. To investigate this, a set of simulations were designed (Set 3 in Table 3) by changing the height of the domain in proportion to the expected $\lambda_{ver}$. Six simulations were run with domain heights in the range of ($\tilde{L}_z = L_z/\lambda_{ver} = 0.25$ to $2.0$) for values of $Fr = 0.1$ and $0.5$. The domain length was set equal to $\lambda_{hor}$ for all simulations, and the damping thickness was set to $1.5\lambda_{ver}$ for $Fr = 0.1$ and $1.2\lambda_{ver}$ for $Fr = 0.5$. Further, $\xi$ was set to 10 for $Fr = 0.1$ and 2.5 for $Fr = 0.5$.

Figure 9b shows the reflection coefficient as a function of the non-damped domain height. A rapid reduction in the reflection coefficient is seen as the domain height is increased and approaches $\lambda_{ver}$. There is a slight increase in the reflection coefficient



for $\tilde{L}_z$ 1.5 for both values of $Fr$, but further increasing the domain height beyond twice $\lambda_{ver}$ has little effect. Our experience indicates that a higher domain height allows more waves to reach the inlet than a lower one. Since the waves are inclined, the wavefronts impinge on the top damping layer in the case of a lower domain before reaching the inlet. However, if the domain height is larger than one $\lambda_{ver}$, then, depending on $Fr$, the wavefronts farther from the source reach the inlet before impinging on the top damping layer. Thus, the reflection pattern seen in Fig. 9b is not entirely monotonic. In any case, it is important to set the minimum height of the non-damped domain height to around one effective vertical wavelength. This recommendation may change when including a temperature profile which more closely reflects an ABL with an inversion layer in which case the inversion height might be critical in setting the non-damped domain height. This will be the subject of further work.

## 5.2 Wind Farm Canopy

The investigation of flow over the hill provided a baseline for simulating AGWs under linearly stratified free atmospheric conditions. This was then extended to the wind farm canopy (WFC). Simulation sets 2 and 3 from Table 3 are re-run, this time with the wind farm canopy. The setups are the same as that of the hill cases described in Section 5.1.5 and Section 5.1.6, except that the hill is replaced with a wind farm canopy where the canopy height and length correspond to the hill's half-width at half-height and maximum height, respectively. The parameters used to model the wind farm canopy are given in Table 4. Similarly to the hill case, the slope parameter is kept constant, i.e., $S_h = 0.16$, and two values of $Fr$, i.e. 0.1 and 0.5, are considered. The thickness of the wind farm layer, i.e., the rotor diameter, the hub height, and $C_t$ are varied as $c_t$ is kept constant, i.e. 0.075. As shown in Table 4, the WFC starts at $H_b = 20\ m$ and goes to $H_t = 80\ m$ vertically for $L = 5\ km$ such that hub height is $h_r = 50\ m$ and and the rotor diameter is $D_r = 60\ m$. For $L = 10\ km$, $H_b = 40\ m$ and $H_t = 160\ m$ with the wind farm layer being $120\ m$ thick and hub height at $100\ m$. Therefore, the turbine thrust is stronger for the $L = 10\ km$ cases than the $L = 5\ km$ cases because of higher hub height and a bigger rotor.

**Table 4.** Wind farm canopy parameters. Note that the canopy only extends over the rotor diameter so the bottom of the canopy is at $H_b$ and the top is at $H_t$.

| Fr | $S_h$ | $c_t$ | $S_x$ | $S_y$ | L (km) | $H_b$ (m) | $H_t$ (m) | $h_r$ (m) | $D_r$ (m) |
|----|-------|-------|-------|-------|--------|-----------|-----------|-----------|-----------|
| 0.1 | 0.16 | 0.075 | 5 | 1.67 | 5 | 20 | 80 | 50 | 60 |
| 0.1 | 0.16 | 0.075 | 2.5 | 0.83 | 10 | 40 | 160 | 100 | 120 |
| 0.5 | 0.16 | 0.075 | 5 | 1.67 | 5 | 20 | 80 | 50 | 60 |
| 0.5 | 0.16 | 0.075 | 2.5 | 0.83 | 10 | 40 | 160 | 100 | 120 |

### 5.2.1 Impact of Damping Layer Thickness

Fig. 10 shows that the results for the canopy case are similar to those of the hill (Fig. 8) in terms of the sensitivity of the reflection coefficient of damping layer thickness and damping coefficient for the two $Fr$ cases. The main difference is that the





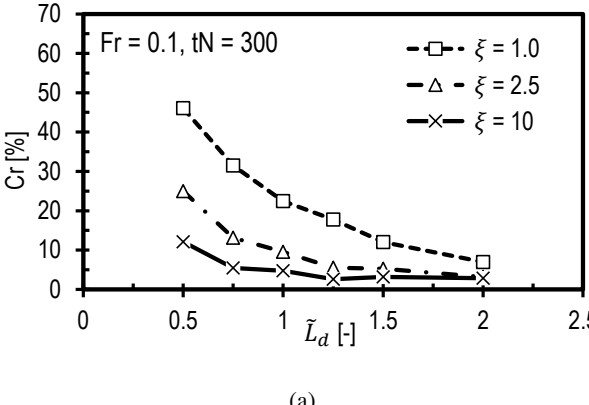
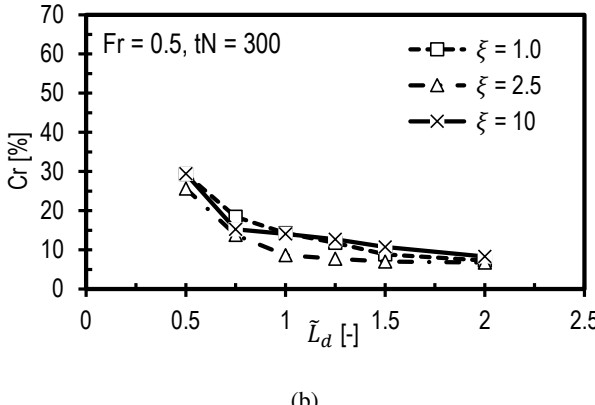

(a)                                                                      (b)

**Figure 10.** Reflection coefficient for the wind farm canopy case as a function of normalised damping layer thickness for different values of the damping coefficient when (a) $Fr = 0.1$ and (b) $Fr = 0.5$.

reduction in $Cr$ with damping layer thickness when $Fr$=0.1 is monotonic for the canopy case and does not show the variation seen for the hill. The reasons for this are explored in Section 5.2.2.

Generally, thicker damping layers compensate for weaker damping coefficients. More importantly, for almost all values of $\xi$ and $\tilde{L}_d < 1$, $Cr$ exceeds the threshold selected in this study for an acceptable level of reflection. Fig. 10b shows that the suitable damping coefficient range for $Fr = 0.1$ is still 1.0 to 10, with 10 being optimal for all damping layer thicknesses. Likewise, Fig. 10b shows that $\xi = 2.5$ is the optimal damping coefficient for $Fr = 0.5$, for all damping layer thicknesses. Furthermore, values of $\xi = 1$ and 10 appear slightly less effective in damping reflections than a value of $\xi = 2.5$.

### 5.2.2    Impact of Domain Size

The domain length is normalized based on the horizontal wavelength predicted from the linear theory for a hill using the wind farm canopy height and length instead of the hill maximum height and half-width at half-height, respectively. We calculate the effective wavelengths from the simulations at $tN = 300$ to compare with the predicted wavelengths where the horizontal wavelength is calculated based on a horizontal slice at $\tilde{L}_z = 1.0$ through the flow fields shown in Fig. 11 and the vertical wavelength is similarly calculated based on a vertical slice at $X = 0$. In general, the calculated wavelengths match the predicted wavelengths.

It is interesting to notice that the wave shape for the canopy is significantly different from that of the hill for the same conditions and optimal simulation setup. The vertical velocity field for the two wind farm canopy lengths and one hill-half width (i.e. $L = 10km$) are shown in Fig. 11. For the WFC, there are two prominent wave trains for the $Fr = 0.1$ cases shown in Fig. 11 (a and c), one at the entrance and the second at the canopy exit. The most dominant is caused by upward flow deflection at the entrance due to the thrust force at the start of the canopy. The wave at the canopy exit results from the downward flow as the thrust force abruptly ends. These waves are out of phase and propagate at the same angle to the horizontal and their interaction leads to a distorted wave spectrum. Therefore, the effective wavelength from a canopy simulation differs from that




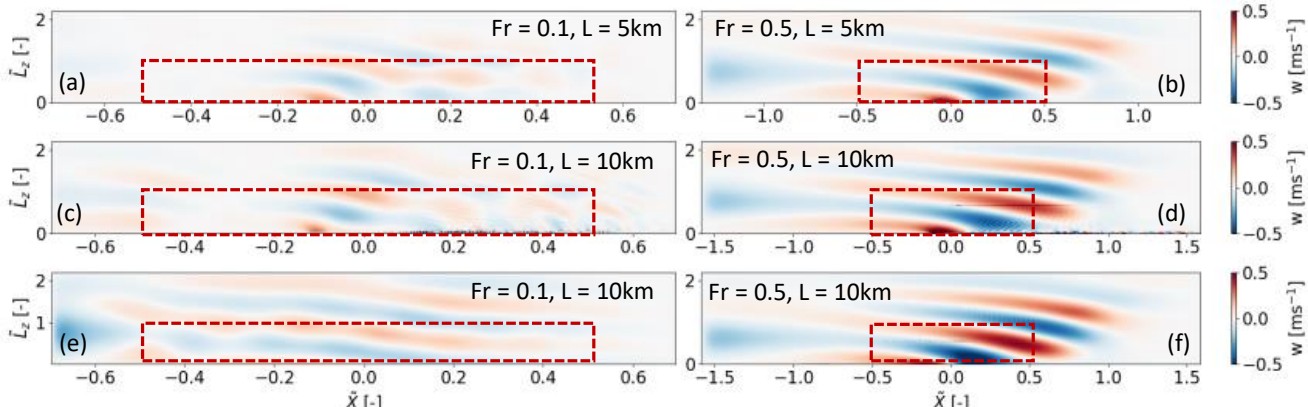

**Figure 11.** Plots (a-d) show vertical velocity of flow through and around the wind farm canopy with lengths $L = 5$ and 10 km for $Fr = 0.1$ and 0.5, and $\tilde{X} = 1.0$. For comparison, the plots (e-f) show the corresponding hill cases where $L = 10$ km for $Fr = 0.1$ and 0.5, and $\tilde{X} = 1.0$. The region inside the dashed red box is the non-damped domain.

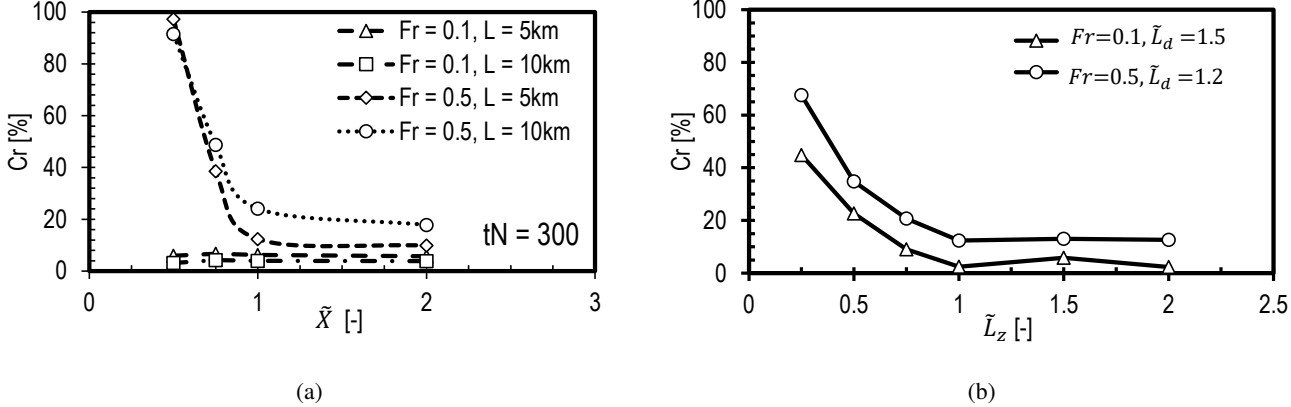

**Figure 12.** Reflection coefficient for the wind farm canopy case as a function of: (a) domain length for $Fr = 0.1$ and $Fr = 0.5$ for $L = 5$ km and 10 km; and (b) domain height for $Fr = 0.1$, $\tilde{L}_d = 1.5$ and $Fr = 0.5$, $\tilde{L}_d = 1.2$.

predicted if the most dominant wave train is considered though the wavelength calculated using the global maxima at the canopy entrance and global minima at the canopy exit does provide a good match. This difference in wave shapes for the WFC and the hill case can be seen by comparing plots in Fig. 11 (a and c with e). When referring to the monotonic $Cr$ plots in Fig. 10a and nearly constant values in Fig. 12a for $Fr = 0.1$ it seems that the dominant wave train triggered by canopy entrance is more critical in terms of simulation setup. For this sub-critical case, this dominant wave train propagates out both upstream and downstream. The second smaller wave train at the exit propagates similarly, merging with the first to give the patterns seen in the plots Fig. 11 (a and c). For $Fr = 0.5$, we see only one wave train in plots in Fig. 11 (b and d) because the advection speeds in this case, 25 and 50 $ms^{-1}$, for the 10 km canopy and the 5 km canopy, respectively, are higher than the wave speed



in this super-critical case. These advection speeds are higher than generally observed for a wind farm but are used here to fix
the value of $Fr = 0.5$ for practical values of $N$ when $L$ is a constraint. As can be seen on plots Fig. 11 (b, d, and f), the wave
shape and wavelengths are the same for the WFC and hill cases when $Fr = 0.5$, however, the amplitude of the vertical velocity
is higher in the hill case.

Another important observation is the increased wave energy accumulation at the inlet in all cases compared with the hill
simulations, suggesting that Dirichlet inflow boundary conditions are inappropriate when simulating AGWs. In contrast to the
zero-gradient outflow boundary condition, which appears to advect the waves through the outlet, the inflow boundary condition
is not influenced by any approaching waves (i.e., the prescribed boundary values remain as prescribed and are not perturbed
by the gravity waves). Thus, the user-prescribed inflow velocity is not truly representative of a flow regime where AGWs are
present. As a consequence, wave energy accumulates at the inlet, and the inlet-RDL can only delay its propagation back into
the domain. Further work concerning how to effectively contain or eliminate this energy accumulation at the inlet is required.

In terms of the main aim of this study, the waves are effectively damped, especially in the top and outlet RDLs, suggesting
that the optimal setups from the hill case are also optimal for the wind farm canopy. Fig. 12a shows that there is no impact of
the domain length on $Cr$ when $Fr = 0.1$. This is because the domain lengths were set as a function of $\lambda_{hor}$ predicted from the
linear theory, which is significantly greater than the wavelength of the dominant wave train at the entrance of canopy. When
$Fr = 0.5$, similar behaviour is seen as for the hill case where $Cr$ reduces with increasing domain length and is minimum when
$\tilde{X} > 1.0$. However, $Cr$ values remain more than $10\%$ because of faster energy accumulation for higher advection speeds than
observed wind speeds for a wind farm.

The impact of domain height on the reflections is also the same as that of the hill case. As shown in Fig. 12b, $Cr$ is minimised
for domain heights greater than $\lambda_{ver}$. It is important to recall that vertical wavelength depends on $U/N$, which holds true for
both canopy and hill cases. The value of $Cr$ is higher when $Fr = 0.5$ because of high amplitudes due to a high advection speed,
i.e. $25 \ ms^{-1}$. Based on these observations, it is suggested to set the domain length and height in wind farm simulations to at
least one predicted effective horizontal and vertical wavelength, respectively. However, the link between high inflow velocities,
wave amplitudes and wave trains should be investigated by modeling wind farms with actuator models for a more accurate
representation of the wind farm flow dynamics.

## 6 Conclusions

This study aims to provide guidelines for atmospheric flow simulations that include atmospheric gravity waves by relating
the key physical and simulation parameters. The study is first carried out for a two-dimensional hill in stably stratified flow
to compare the results to an analytical solution. The findings are then tested for flow through wind farms, approximated with
a wind farm canopy model. Based on recent findings in the literature, only Rayleigh damping gravity wave treatment was
investigated. Therefore, the findings apply to simulation setups with Rayleigh damping layers and inflow/outflow boundary
conditions solved with finite-volume codes.





Simulation time is one of the most critical parameters in simulations, including gravity waves. In all cases, longer simulation time would result in the accumulation of wave energy at the inlet boundary, and the reflections gradually become stronger. Thus, we conclude that the Rayleigh damping method attenuates the gravity waves to an extent that may not work for a long simulation, such as a diurnal simulation. Therefore, a robust technique is required to handle both the energy accumulation and reflections.

The results regarding the configuration of the damping layers show a trade-off between the ability to correctly resolve gravity waves and computational resources. With periodic conditions in the lateral direction, the highest accuracy can be achieved with damping layers of a thickness exceeding the effective vertical wavelength at the inlet, top, and outlet. In case of limitations on the computational resources, a combination of damping layers of the same thickness at the inlet and top could still be reasonable.

Our test shows that for various Froude numbers, small damping coefficients would not be effective even for damping thickness 2 to 5 times greater than the effective vertical wavelength. Likewise, the reflections are higher for layers with excessively large damping coefficients and might distort the solution in regions of the non-damped domain close to the damping layer. The most suitable damping coefficients are values from 1 to 10 when the damping coefficient is normalized with Brunt-Väisälä frequency. The thickness of the damping layer should be at least one effective vertical wavelength, and thicknesses exceeding 1.5 times the effective vertical wavelength may be unnecessary.

The domain length should be scaled with the effective horizontal wavelength and not with the length of the hill or wind farm. The reflection coefficient as a function of domain length normalized by effective horizontal wavelength shows that large domains are more appropriate to avoid undesirable levels of reflections from the domain boundaries. For domain lengths exceeding one horizontal wavelength, the reflection of the upward propagating energy is less than 6% in the hill case. The reflection coefficient was slightly higher for the wind farm canopy cases due to the interaction of the wave trains at the canopy entrance and exit. Therefore, further investigation using such as an actuator disk approach to model the wind turbines is required to thoroughly test the optimal setups and understand the wave dynamics. For both flow scenarios, i.e. the hill case and the canopy case, increasing the domain length beyond $\lambda_{hor}$ shows a small reduction in the reflection coefficient. Similar impact on the reflection coefficient is observed when varying the non-damped domain height, and setting it to at least one effective vertical wavelength is recommended. It is important to point out that the wave reflection coefficient in this study was calculated by taking the ratio of energy values. This can be misleading in some cases, as the reflected waves can be directed upwards, thus reducing the $Cr$ value. Visual inspection of the vertical velocity fields is recommended to cross-check the $Cr$ values.

These recommendations are based on linearly stratified atmospheric flows. Aspects like turbulence and a more realistic and complex temperature structure of the atmosphere are not considered in this study. Nevertheless, this work should help provide useful guidelines for setting up simulations that include atmospheric gravity waves in wind energy applications. We have tested these findings with another finite-volume code, TOSCA (Stipa et al., 2023), and found consistent results with SOWFA.

In further work, the impact of the inversion layer (inversion Froude number and height) on the setup will be explored as will the impact of turbulence and the Coriolis force. More importantly, it is intended to develop an inflow boundary condition that can avoid the energy accumulation at the inlet based on an inflow which accounts for the presence of AGWs.





*Data availability.* The raw data from LES and post-processing scripts will be archived as a repository on the 4TU drive of TU Delft and will be open access.

*Video supplement.* Videos can be made available to highlight the time evolution of the IGWs for both flow scenarios addressed in this study.

## Appendix A: Configuration of Damping Layers

605 The reflection of waves from the top boundary is expected as the typically used boundary conditions to model an arbitrary location in the atmosphere are mostly fully reflective. Since gravity waves travel vertically, one would anticipate reflections from the top boundary if forcing zones are not used there. Intriguingly, with inflow/outflow boundary conditions, the damping layers are needed at other boundaries, too. For instance, the reflected waves from the top are directed towards the outlet, and weak reflections may also appear there. Likewise, the wave energy distributes throughout the domain and accumulates over 610 time if not damped at the boundaries. This is evident as reflections from the inlet propagate into the domain after one to two flow-through times. Thus, damping the waves at various boundaries is necessary for realistic simulations. Sometimes, the solution for a setup without an appropriate damping layer configuration will diverge due to numerical instability caused by excessive flow velocity from complex reflected wave interactions.

In this context, deciding the configuration of damping layers becomes important when setting up simulations involving 615 gravity waves. We explored the configuration of damping layers to come up with a base case. The test cases and normalised errors are shown in Fig. A1.

At this point, it is important to highlight that the minimum amount of reflections is a user choice. Since the reflections intensify over simulation time, it may be the case that a user will opt for a configuration based on the availability of computational resources and the simulation time of their interest. Thus, a simulation without any forcing zones is the first possibility. The 620 domain height is critical when there is no damping. This can be established by comparing cases 1, 2, and 3, where the domain height ($L_z$) is 1, 2, and 3 times the effective vertical wavelength, respectively. Reflections increase rapidly when increasing the domain height, from 22% to 33% and 120% in cases 1 to 3, respectively. However, none of these cases fall below the criterion for an acceptable amount of reflection. Thus, we opt to use Rayleigh damping, and it is evident that only having a top Rayleigh-damped layer (case 4) is insufficient to reduce reflections to an acceptable level. The R-RMSE for this case is 625 39%, and the solution for the hill case is contaminated by reflections from the inlet such that the actual gravity waves disappear entirely. This suggests that it is better not to have a damping layer rather than only having a damping layer at the top and instead have a setup with a domain height just over one effective vertical wavelength. This would be viable only when a maximum value of R-RMSE or Cr of about 20% is an acceptable criterion for a given problem.

The most suitable set up is to have damping layers at the inlet, outlet, and top with the same damping characteristics. This 630 is Case 5 in Fig. A1, where R-RMSE is only 2.38%. If the damping layer thickness is reduced by half whilst having damping on all three sides (Case 6), the R-RMSE increases to 10%. A combination of damping layers at the top and outlet (Case 7)

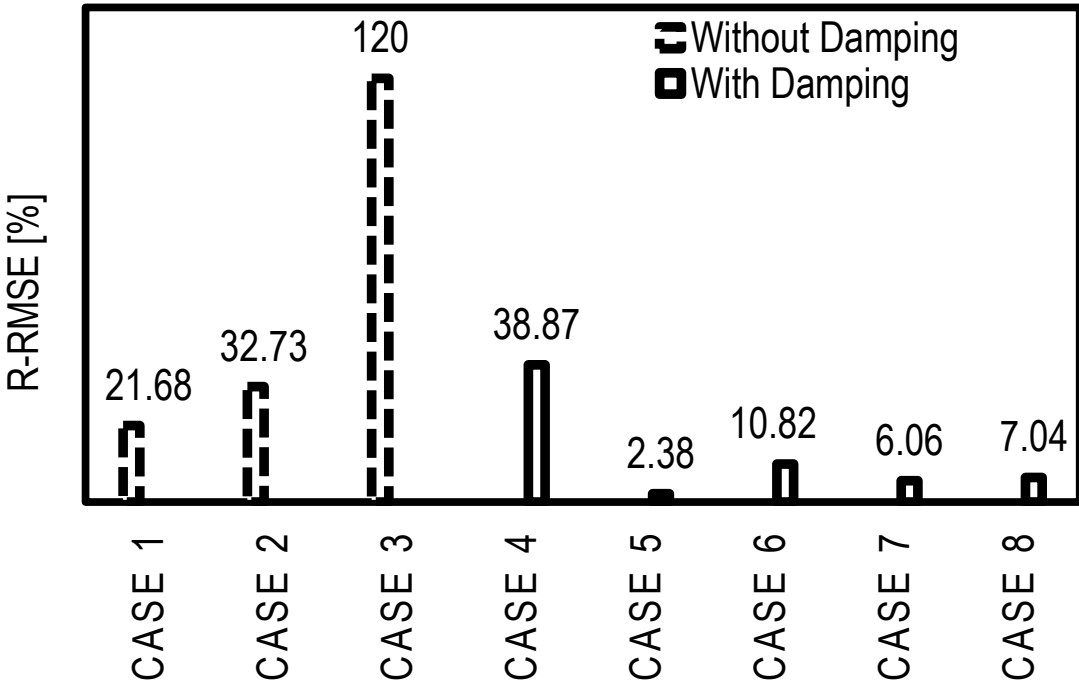

**Figure A1.** Comparison of R-RMSE for various damping configurations designed to acquire the most suitable arrangement for the base case.

appears better than Case 6 as R-RMSE here is 6%. Damping at the top and inlet (Case 8), is slightly more reflective (R-RMSE 7%) than in Case 7. There can be other possible configurations, but this analysis gives enough insight to decide on a suitable configuration for a sufficiently accurate base case. Altogether, the configuration of damping layers is a trade-off between
computational resources and the desired accuracy of the solution that depends on the user's choice regarding the acceptable degree of reflection.

*Author contributions.* Conceptualization, M.A.K, D.A.; methodology, M.A.K, D.A; software, M.A.K; validation, M.A.K; formal analysis, M.A.K; investigation, M.A.K, D.A; computational resources, M.A.K, D.A; data curation, M.A.K; writing–original draft preparation, M.A.K; writing–review and editing, D.A, S.J.W, M.J.C.; visualization, M.A.K; supervision, D.A, S.J.W, M.J.C.; project administration, D.A, S.J.W;
funding acquisition, D.A, S.J.W.. All authors have read and agreed to the published version of the manuscript.

*Competing interests.* No competing interests are present.



*Acknowledgements.* This publication is part of the project: Numerical modelling of Regional-Scale Wind Farm Flow Dynamics, with project number: 2023/ENW/01454045 of the research programme: ENW which is (partly) financed by the Dutch Research Council (NWO).

R





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
