# Peer review of "Investigating the Relationship between Simulation Parameters and Flow Variables in Simulating Atmospheric Gravity Waves for Wind Energy Applications"

_Wind Energy Science, 2024_

## Referee Comment (RC1)

**Review of the manuscript wes-2014-138 entitled "Investigating the Relationship between Simulation Parameters and Flow Variables in Simulating Atmospheric Gravity Waves for Wind Energy Applications" by M. A. Khan, D. Allaerts, S. J. Watson, M. J. Churchfield**

**Overview**

This study addresses the problem of damping the atmospheric gravity waves (AGW) spuriously reflecting from the boundaries of a numerically simulated domain depending on the provided boundary conditions. Great attention is devoted to the choice of the numerical simulation setup (i.e. domain length and height) in relation to the main non-dimensional parameter governing the problem (i.e. Froude number and aspect ratio of the obstacle triggering the waves). The results, arising both from the simulation of the flow around a hill and a wind farm, indicate that, for low Froude numbers (typical of wind farms operating in a stably stratified flow), the horizontal and vertical AGW represent the best way to effectively scale the simulation domain to minimize the reflection of gravity waves. Furthermore, when a Rayleigh damping layer is used, the optimal damping coefficient scales with the Brunt-Vaisala frequency.

The article is well written and addresses an important topic for the wind energy community. Thus, I recommend its publication on WES after some minor corrections I reported below.

**General comments**

- **Line 207**: For the sake of clarity, I would write the steady-state, semi-analytical solution explicitly.

- **Line 215**: Is there a more rigorous way to define the dominant wavelength than the most apparent in flow visualizations?

- **Lines 229-230**: From Fig. 2, it seems like the energy associated to $\lambda_{hor}$ is not constant as the streamwise coordinate increases. Does this quantity depend on the length of the streamwise domain itself?

- **Line 232**: As per the steady-state solution, I would write the explicit form of the amplitude as a function of the wavenumber. This would help understanding the following points the authors made.

- **Line 241**: By mentioning the vertical wavelength, do you imply to take the FFT along the vertical direction? The $z$-coordinate is not homogeneous (due to the Dirichlet condition typically imposed at $z = 0$), so this operation cannot be done without further clarifications.

- **Line 299**: For the wind farm case, did you provide a velocity profile at the outlet, or did you implement a Neumann boundary condition?

- **Line 337-338**: The definition of upward and downward moving waves is unclear. The quadrant definition depends on how you define positive and negative wave number coordinates, which is unclear as of now. Please further clarify this aspect or add a figure to better explain it.

- **Lines 483-486**: Besides the effects of domain length and height, I would also mention the AGW's angle among the factors determining the reflection coefficient, at least reporting the interval of observed AGW angles.

**Technical corrections**

- **Line 102**: When you mention the "sponge zones", please put the comma outside of the quotes.

- **Figure 3b**: The representation of the main wavenumbers of interest is a bit confusing when all of them are reported with the same color. Besides using different line styles, I recommend using different colors as well.

- **Line 236**: Please replace "then" with "than".

- **Line 283**: I would put a comma between "$Fr$" and "because" instead of a period.

- **Line 293**: Please remove either "used" or "solves" as they sound redundant.

- **Line 400**: Please replace "seem" with "seen".

- **Line 598**: Please replace "as will" with "as well as".

---

## Author Comment (AC1)

**Delft University of Technology, Netherlands**

This study is part-1 of the project:
"Numerical Modelling of Regional-Scale Wind Farm Flow Dynamics"

**Response to Reviewer 1**

Exec: M.A.Khan - February 18, 2025
Revised: Simon Watson - February 18, 2025
Revised: Matthew Churchfield - February 18, 2025

The authors thank the reviewer for the time dedicated to revising the article and suggesting valuable feedback to improve this research and the article. We proceed with answering and clarifying their comments, where possible.

For ease of tracking, our response to each comment is denoted in black, while the reviewer's comments are denoted in blue. The detailed changes made to the manuscript are included in the "track changes" section at the end of this document.

Line 207: For the sake of clarity, I would write the steady-state, semi-analytical solution explicitly.

We agree with the reviewer, and this line now reads, "The steady-state, semi-analytical solution of uniform flow over the hill..."

Line 215: Is there a more rigorous way to define the dominant wavelength than the most apparent in flow visualizations?

Following the suggestion of the reviewer, we include the expression for the dominant wavelength, and the modified lines read: "The wavelength with the highest amplitude is the *dominant* wavelength, $\lambda_L = 2\pi L$, which depends on the hill width."

Lines 229-230: From Fig. 2, it seems like the energy associated to $\lambda_{hor}$ is not constant as the streamwise coordinate increases. Does this quantity depend on the length of the streamwise domain itself?

We do not fully understand your question. We think by "energy," you are referring to the wave amplitude, and you are referring to the fact that wave amplitude is greatest within a band of the streamwise coordinate near and just downstream of the hill. Further downstream of the hill, this wave amplitude decreases. When you ask, "does this quantity depend on the length of the streamwise domain itself?" do you mean: Does the wave amplitude depend on streamwise location? Or do you mean: Does the distribution of wave ampltitude in the streamwise direction depend upon domain streamwise length?

Yes, the wave amplitude depends on streamwise location. A mathematical way to view this is that the waves originate from the flow displacement caused by the hill. This flow displacement in physical space can be transformed to wavenumber space and contains the full range of frequencies ranging from the largest being the length of the domain to the smallest being much smaller than the hill. The buoyancy frequency (i.e., the time-based frequency at which an air parcel would bob up and down if displaced in this stable flow) combined with the mean wind speed (imagine the path traced out by an air parcel bobbing up and down and blowing along) can only support wavenumbers up to a cutoff wavenumber. The higher the wind speed, the lower the cutoff wavenumber; the lower the buoyancy frequency, the lower the cutoff wavenumber. Wavenumbers above the cutoff wavenumber decay exponentially with height above the surface whereas the wavenumbers below the cutoff remain. Therefore, we see the aspect of wave amplitude decaying with height. And, when we transform this truncated wavenumber representation back into physical space, we see waves that have amplitude that decay beyond the hill due to the truncation of high wavenumbers. A physical way to view this is that "the gravity waves transport energy mainly upward from their source, the hill or a wind farm, but some wave amplitude decays with height. At the same time, the waves are also blown downstream. The combined effect is a decrease in the amplitude of some parts of the wave with height and downstream distance."

Does all of this depend on domain length? Yes, if the domain length is too short, boundary effects will affect the solution. However, we use very long domains that we know, through previous experience, are long enough to avoid spurious boundary effects.

Line 232: As per the steady-state solution, I would write the explicit form of the amplitude as a function of the wavenumber. This would help understanding the following points the authors made.

This valuable suggestion of the reviewer is implemented by including the expression for amplitude and relevant commentary. Furthermore, we have rearranged the paragraphs, lines 207-266, in the "Semi-Analytical Solution" section to improve the discussion and coherence. This change can be seen in the "tracked changes" section at the end of this document.

Line 241: By mentioning the vertical wavelength, do you imply to take the FFT along the vertical direction? The z-coordinate is not homogeneous (due to the Dirichlet condition typically imposed at z = 0), so this operation cannot be done without further clarification.

This is well-noticed by the reviewer, and we clarify the definition of vertical wavelength by indicating it in Fig. 2a and including Fig. 3b, which shows the normalized effective vertical wavelength as a function of $Fr$. To be explicit, we do not perform FFT. Instead, we computed the vertical wavelength from the vertical velocity in the physical coordinates. The definition of $\lambda_{ver}$ and the commentary on these figures are given below.
"The effective vertical wavelength ($\lambda_{ver}$), shown as the red vertical arrow in Fig. 2a, depends only on the Scorer parameter, which is proportional to the vertical wave number. This is evident from Fig. 3b, where $\lambda_{ver}$ normalized with the maximum vertical wavelength, $2\pi(U/N)$, is shown as a function of $Fr$. The constant $\lambda_{ver}/2\pi(U/N)$ for varying $Fr$ indicates that $\lambda_{ver}$ does not depend on $L$ and is usually representative of the maximum vertical wavelength. $\lambda_{ver}$ is measured as twice the distance between the global maxima and minima determined from a vertical velocity plot along the domain height at the x-location corresponding to $H$. The locations of global extrema are extracted from the vertical velocity on the physical coordinates such that grid stretching does not affect $\lambda_{ver}$. It is also important to note that the IGWs curve downstream of the hill; thus, $\lambda_{ver}$ slightly varies at different streamwise locations for values of $Fr > 0.5$ as seen in . 3b. Therefore, the maximum vertical wavelength is better suited to scaling in the vertical direction, especially since it is almost equal to $\lambda_{ver}$

Line 299: For the wind farm case, did you provide a velocity profile at the outlet, or did you implement a Neumann boundary condition?

We implement the zero-gradient Neumann boundary condition at the outlet for velocity in both flow scenarios. This is now explicitly mentioned in line 321 of the revised manuscript.

Line 337-338: The definition of upward and downward moving waves is unclear. The quadrant definition depends on how you define positive and negative wave number coordinates, which is unclear as of now. Please further clarify this aspect or add a figure to better explain it.

We have tried to better explain the process of filtering upward and downward moving waves in calculating $Cr$, and included Fig. 5 that shows the decomposed waves in vertical velocity fields. The revised description in lines 351-359 of the modified manuscript read as follows:
"The reflection coefficient ($Cr$) is one of the two primary tools used here to analyze the simulation data, and its calculation can be summarised in the following two steps. First, the vertical velocity values on a vertical streamwise plane are converted to horizontal and vertical wave number space ($K_s$ and $M_s$, respectively) through a 2D Fourier transform. The wave number coordinates centered at 0 are similar to the physical coordinates where $K_s$ and $M_s$ are on the horizontal and vertical axis, respectively. The upward and downward moving waves are separated based on the quadrants they fall into. The upward propagating waves are in the first and third quadrants, whereas the downward propagating are in the second and fourth quadrants on the $K_s$ and $M_s$ coordinates. Thus, only the Fourier coefficients in quadrants I (i.e., $K_s, M_s > 0$) and III (i.e., $K_s, M_s < 0$) are retained for upward waves, and those in the other two quadrants are set to zero.

Likewise, quadrants II (i.e., $K_s < 0, M_s > 0$) and IV (i.e., $K_s > 0, M_s < 0$) are retained for the downward waves. The filtered coefficients are inverse-transformed to obtain the respective velocity fields, as shown in Fig. 5. The tilted phase fronts on these contours clearly show (a) the initial upward, and (b) the much weaker reflected downward propagating waves. Wave energy propagates normal to the phase fronts and is calculated from these decomposed vertical velocity fields. Finally, the reflection coefficient is estimated by taking the ratio of total downward to upward propagating energy."

Lines 483-486: Besides the effects of domain length and height, I would also mention the AGW's angle among the factors determining the reflection coefficient, at least reporting the interval of observed AGW angles.

This is an important suggestion from the reviewer as the inclination angle of the AGWs is critical but complicated to analyze, especially in the wind farm canopy scenario. Thus, we have included the inclination angles observed for the range of $Fr$ that we investigate in this study for the hill scenario. This suggestion is addressed in lines 231-235 of the revised manuscript that read:
"Although the analytical solution tells us that wave amplitude depends more on the hill slope, particularly the height, the effective wavelengths and the inclination angles of the IGWs depend on $Fr$. The IGWs naturally tend to travel vertically, but the background flow forces them to bend. Since zero advection speed is impossible, these waves would always travel inclined. For $Fr$ in the range of 0.1 to 1.5, the inclination angle to streamwise direction, estimated from vertical velocity field, ranges $75 - 53^o$, respectively.".
Moreover, the inclination angle does not affect the wave damping because Rayleigh damping is implemented on the velocity components. This is explicitly mentioned in the conclusions of the new manuscript, which reads: " The wave direction is critical to understanding AGWs, but it should not affect the wave damping as it is applied to velocity components; usually, the vertical velocity is set to zero, and the other two are set to the geostrophic components.".

Line 102: When you mention the "sponge zones", please put the comma outside of the quotes.

This technical comment is implemented as suggested by the reviewer.

Figure 3b: The representation of the main wavenumbers of interest is a bit confusing when all of them are reported with the same color. Besides using different line styles, I recommend using different colors as well.

This technical comment is implemented as suggested by the reviewer, please see Fig 2b in the manuscript showing tracked changes.

Line 236: Please replace "then" with "than".

This technical comment is implemented as suggested by the reviewer.

Line 283: I would put a comma between "Fr" and "because" instead of a period.

This technical comment is implemented as suggested by the reviewer.

Line 293: Please remove either "used" or "solves" as they sound redundant.

This technical comment is implemented as suggested by the reviewer.

Line 400: Please replace "seem" with "seen".

This technical comment is implemented as suggested by the reviewer.

Line 598: Please replace "as will" with "as well as".

This technical comment is implemented as suggested by the reviewer.

**References**
**1 Track Changes**

In the following we report the changes made to the paper using latexdiff.

[revised manuscript text omitted]

---

## Author Comment (AC2)

**Delft University of Technology, Netherlands**

This study is part-1 of the project:
"Numerical Modelling of Regional-Scale Wind Farm Flow Dynamics"

**Response to Reviewer 2**

Exec: M.A.Khan - February 18, 2025
Revised: Simon Watson - February 18, 2025
Revised: Matthew Churchfield - February 18, 2025

The authors thank the reviewer for the time dedicated to revising the article and suggesting valuable feedback to improve this research and the article. We proceed with answering and clarifying their comments, where possible.

For ease of tracking, our response to each comment is denoted in black, while the reviewer's comments are denoted in blue. The detailed changes made to the manuscript are included in the "track changes" section at the end of this document.

What is role of grid resolution in the conclusions? Certainly, the domain size conclusions will be unchanged, but is the damping strength dependent on grid resolution?

For the investigated parameter space (i.e., $Fr$ between 0.1 to 0.5), the grid resolution used in the study does not affect the simulation setup. This is because the AGW wavelengths, typically 3 to 45 $km$ are many, many times bigger than the typical grid sizes, 10 to 20 $m$, used in wind farm LES (the ultimate objective of this research). Since we are fully resolving the AGWs using a resolution typical of LES, the solution is grid-independent. Rayleigh damping contains no length scale, so it is not dependent on grid resolution. This reasoning is mentioned in the text in lines 304-305 and is now highlighted in the conclusions as follows:
"In terms of time-step size and grid resolution, the typical values used in wind farm LES (i.e., $0.5 < t < 1.0$ and $5 < \Delta x, \Delta y, \Delta z < 20$) are more than sufficient to resolve the AGWs. Thus, the simulation setups used for the investigated $Fr$ range are independent of the time-step size and grid resolution because the time periods and wavelengths of AGWs are several times bigger."

Why are the simulations three-dimensional? There are only five grid points in the third dimension without any inherent three-dimensionality in the geometry or inflow, so the flow is almost certainly two-dimensional.

It is true that the flow is two-dimensional and including the third dimension was not necessary. However, the wind farm canopy case triggers turbulence. Thus, we considered three-dimensional simulations to resolve at least some turbulence induced by the canopy. For consistency, we used the setup in the hill scenario. Also, we were unsure if flow separation would occur downstream of the hill, which didn't because the slope parameter was constant.

In various places in the manuscript, the authors mix discussion of dimensional terms and non-dimensional terms. Everything seems to be able to be expressed in non-dimensional terms, and I encourage the authors to rewrite the commentary and replot the figures as such where possible. Doing so could also reduce the total number of figures and lines on figures needed. (The need to demonstrate dynamic similarity also seems unnecessary.)

It is correct that every plot and discussion can be expressed in non-dimensional terms; in fact, the plots are all in terms of non-dimensional numbers. To provide additional insight, the plots related to domain length include hill-half widths or canopy lengths, this is indeed redundant as $Fr$ already includes $L$. Thus, we have revised the plots in Fig. 10a and Fig. 13a by keeping only $Fr$.
However, we would like to keep the vertical velocity contours in Fig. 12 and the commentary on the same as its visual evidence of AGW-dependency on $L$ for low values of $Fr$ and on $U/N$ for higher values of $Fr$. Likewise, the plots in Fig. 6 about dynamic similarity and the related discussion are important in introducing critical length and time scales in studying AGWs. Further, such representation helps in interpreting the physical meaning of the non-dimensional number, i.e. $Fr$. Such as, "Thus, it is deduced that by the definition $Fr = U/NL$, $L/U$ acts like $N$ or the free atmosphere's stability. This means that the stability of the free atmosphere is relative to the size of the disturbance source and the geostrophic wind."

A key conclusion of the manuscript is the need to scale the domain size and damping layer dimensions by

the wavelengths. However, the actual discussion of the wavelengths from the semi-analytical solution for the hill is somewhat limited. The vertical wavelengths are not actually quantified or plotted, in fact. To make the manuscript more useful, the authors should include expressions for the (non-dimensionalized) wavelengths or similar plots as a function of the relevant non-dimensional parameters. That will allow other researchers to consult this work in order to scale their geometries.

The reviewer has rightly indicated the need to include further discussion on the vertical wavelength. As suggested by them, we have included a plot in Fig. 3(b) of the revised manuscript that shows non-dimensional vertical wavelength as a function of $Fr$. Please check Fig. 3(b) and commentary on it in the "tracked changes" section.

While the authors allude to other effects of interest (turbulence, ABL, etc.), for future work, this leaves the reader wondering how relevant this work really is. For example, no wind farm is strictly in the free atmosphere, and this seems to be intended application of the authors. How then would the ABL influence the conclusions of this manuscript? While the authors may not have the quantitative answer yet, some scaling argument to demonstrate that the present results would, for example, be a conservative constraint on more complex flow situations would be beneficial.

There are two points to note in the reviewer's comment: first, how relevant is this study in modeling wind farms, and second, how the conclusions of this study may vary when an ABL is included? Firstly, this study is focused on the most important aspect of modeling and understanding wind farm-induced atmospheric gravity waves, which is that the internal and interfacial waves are the most critical ones in the accurate modeling of AGWs, and they occur in the free atmosphere. Moreover, these are the waves that reflect from the domain boundaries, and damping layers are always implemented only in the free atmosphere region of the simulation domain. Thus, focusing on the free atmosphere is the initial and most important piece of the puzzle and by only clearly understanding it, one can piece together the remaining. Altogether, we are confident in proposing this study as the stepping stone to accurately modeling and investigating AGWs. These aspects are already mentioned in the manuscript, such as, "It is critical to note that RDLs are always implemented in the free atmosphere because it is the gravity waves in the free atmosphere that reflect from the boundaries".

For the second question, the findings of this study are already validated for CNBL conditions by the authors in a follow-up study Khan et al. (2024). We have mentioned this in the conclusion besides citing the follow-up study for the readers to consult. Thus, the last paragraph of the conclusions section is modified as follows.

[revised manuscript text omitted]